# Hierarchical organization and assembly of the archaeal cell sheath from an amyloid-like protein

Hui Wang [1,2,3], Jiayan Zhang[2,3], Daniel Toso[1,2,3], Shiqing Liao[2,3], Farzaneh Sedighian[3], Robert Gunsalus[3,4] & Z. Hong Zhou [1,2,3] ✉

Certain archaeal cells possess external proteinaceous sheath, whose structure and organization are both unknown. By cellular cryogenic electron tomography (cryoET), here we have determined sheath organization of the prototypical archaeon, *Methanospirillum hungatei*. Fitting of Alphafold-predicted model of the sheath protein (SH) monomer into the 7.9 Å-resolution structure reveals that the sheath cylinder consists of axially stacked β-hoops, each of which is comprised of two to six 400 nm-diameter rings of β-strand arches (β-rings). With both similarities to and differences from amyloid cross-β fibril architecture, each β-ring contains two giant β-sheets contributed by ~450 SH monomers that entirely encircle the outer circumference of the cell. Tomograms of immature cells suggest models of sheath biogenesis: oligomerization of SH monomers into β-ring precursors after their membrane-proximal cytoplasmic synthesis, followed by translocation through the unplugged end of a dividing cell, and insertion of nascent β-hoops into the immature sheath cylinder at the junction of two daughter cells.

Essential to life emergence, the barriers separating the interior of a cell from the exterior differ significantly among the three domains of life. While all cells possess lipid bilayers that enclose their cytoplasm, some types specialize by assembling additional exterior layers that confer shape and size. For example, plant cells possess a cellulose-based outer layer and most bacteria have peptidoglycan layer(s). Some bacteria and most archaea have an S-layer. A distinct group of microorganisms possess an additional outer layer, termed the sheath, with the best-studied member of this group being the methanogenic archaeal species, *Methanospirillum hungatei*. First isolated from a municipal sewage sludge digester, it operates in combination with fermentative and syntrophic bacteria to recycle complex organic matter into methane, water, and carbon dioxide[1]. It provides the driving force for carbon decomposition in anaerobic food chains and thus performs an essential role in the carbon cycling on earth[2]. *M. hungatei*, together with other specialized methanogens, possess an unusual cylindrical cell morphology[3,4]. Its cylindrical shape is determined by an outermost proteinaceous sheath layer that ensures a uniform cell dimension of 0.4–0.5 μm in diameter[3,5] and encapsulates from 1 to over 70 cells within, forming a slightly wavy filament up to 500 μm in length[1,6]. Within the sheath tube, individual cells are enclosed by a proteinaceous surface layer (S-layer) outside their lipid membrane and further separated by plug-like structures. The polar tufts of flagella and pili at the two ends of the sheath tube allow for cell motility and taxis[1,4,5]. As a hydrogenotrophic methanogen, *M. hungatei* is both a consumer and producer of gases, resulting in differential internal pressures relative to the outside environment. Its cylindrical shape enhances gas exchange, given that a representative *M. hungatei* cell that is ~7 μm in length[7] and 0.4 μm in diameter, its sheath can double the surface area compared to that of a sphere shape with the same volume. Since the first observations of these cells nearly half a century ago[1,5], negative-stain transmission electron microscopy (TEM) has defined the basic cell

[1]Department of Bioengineering, University of California, Los Angeles (UCLA), Los Angeles, CA 90095, USA. [2]California NanoSystems Institute, UCLA, Los Angeles, CA 90095, USA. [3]Department of Microbiology, Immunology, and Molecular Genetics, UCLA, Los Angeles, CA 90095, USA. [4]The UCLA-DOE Institute, UCLA, Los Angeles, CA 90095, USA. ✉e-mail: Hong.Zhou@UCLA.edu

architecture of *M. hungatei*, both for the whole cell and cell fraction[5,8], as well as the organization of the plug, S-layer lattices and sheath[9–12]. The sheath is an extremely stable structure and maintains the cylindrical shape of the cell under common denaturants[13]. Early attempts to isolate the sheath protein (SH) under strongly reducing conditions first yielded flattened crystalline arrays[8] and subsequently multiple hoops[14,15], suggesting that the subunits of the cylindrical sheath layer do not follow a helical arrangement. Recent results from a combination of conventional TEM, mass spectrometry, and sequence analysis establish that SH is the 40.6 kDa protein WP_011449234.1 previously annotated from the *M. hungatei* genome and suggested to share amyloid properties[16].

However, molecular descriptions of the sheath are lacking. Indeed, *M. hungatei* sheath's diverse, and seemingly unrelated characteristics (single protein composition, large cylindrical shape, resistance to high pressure[17], hydrogen–methane gas exchange, etc.) raise more questions than answers. What is the structure of the sheath protein? How does it assemble and maintain the characteristic cylindrical shape of the cell to permit gas exchange and direct cell growth? How are the newly synthesized SH translocated through multiple barriers to assemble into well-organized polymers outside the cell? Here, by cellular cryogenic electron tomography (cryoET) with subtomogram averaging and modeling with SH monomer from AlphaFold[18], we establish the molecular organization and biogenesis of the cylindrical sheath layer of *M. hungatei*. About 450 SH monomers circumferentially polymerized around the cylindrical cell. They form a ring structure (termed as β-ring), which consists of two enormous β-sheets with characteristics similar to the so-called "β-arch kernel" coined for amyloids[19]. β-hoops, each comprising 2–6 β-rings, stack axially into a cylindrical sheath extending to hundreds of micrometers long. In addition, visualization of SH structures with an immature cell allows us to propose a route for nascent β-hoop synthesis, assembly, and insertion into a growing sheath, answering the question of how proteins are transported across multiple barriers to assemble into the largest known assembly from a single protein—the external sheath of archaeal cells.

## Results

### Organization of the *M. hungatei* sheath by cellular cryoET

To examine *M. hungatei* cells and resolve its sheath structure at higher resolution and without dehydration and fixation artifacts, we captured in situ cell images by cellular cryoET (Fig. 1). Lower magnification tomograms reveal the overall cylindrical shape of the sheath layer with ~400 nm diameter, as well as organelle distribution within the cell (Fig. 1a, b). Cross-section views of higher magnification tomograms (Fig. 1c) show that the sheath is composed of a series of hoops, which we refer to as β-hoops, to reflect their composition of β-sheet-rich SH (see below). Thousands of β-hoops stack axially to form a tube-like structure that encases cells, whose topology and "crystalline" arrangement have been examined at low resolution by cryoET[3] and negative-stain TEM[14,15], respectively.

Our high-resolution tomograms now show that the sheath actually is a non-periodic stack of β-hoops with variable numbers of SH rings (β-rings), thus deviating from the previously thought crystalline arrangement (Fig. 1c). The cross-section of each β-hoop has a hand-like appearance (Fig. 1c insets) with average "hand" length at around 9 nm. However, the number of "fingers" of the "hand" varies from 2 to 6, which results in the width variation of β-hoops (Fig. 1c and Supplementary Fig. 1a), thus the previous TEM-based interpretation of sheath being crystalline[14] is actually incorrect, explaining the failure of the early efforts, including our own, to resolve the sheath structure based on crystallinity. Each of these "fingers" is the cross-section view of a β-ring. The predominant β-hoop type is the one containing 4 β-rings, while β-hoops containing 3 or 5 β-rings occur less frequently, and those with 2 or 6 β-rings rarely (Fig. 1d and

Supplementary Fig. 1a; termed 2-, and 6-β-ring hoops, respectively). As indicated by the arrangement of different membered β-hoops, no long-range periodicity is observed along the axial direction. In addition, this pattern at the opposite side of the cell is not always maintained, indicating that β-rings in adjacent hoops can switch memberships (Supplementary Fig. 2).

To further improve resolution and investigate the building block of the sheath layer, we classified subtomograms into different membered β-hoops. By averaging β-hoops with the same number of β-rings, we obtained subtomogram averages and resolved the 4-β-ring hoop at 7.9 Å global resolution (anisotropically varying from 7 to 20 Å at different directions) (Supplementary Fig. 1b, c and Supplementary Movie 1). Placing this subtomogram average map back into the original tomogram allowed us to generate a large region of the sheath, containing 3 full β-hoops and 2 half β-hoops (hoops colored differently in Fig. 1e and Supplementary Movie 2), which furthers structural understanding of the sheath within its in situ environment. The structural feature appears to repeat every 28 Å circumferentially along the β-ring, consistent with previous observation[14] (Fig. 1f, repeating units are demarcated by a dashed line of the same color). 3D segmentation was conducted to visualize this repeating unit (Fig. 1g), which matches the AlphaFold-predicted model of an SH monomer (Fig. 1h). This SH monomer model can be divided into three domains: a cap domain containing six α helices and one β sheet, an amyloid-like domain containing two six-stranded large β sheets, named β-sheet 1 and β-sheet 2, and a linker projecting from the joint between the cap and amyloid-like domain (Fig. 1g, h). A portion of a β-ring was modeled by fitting the AlphaFold-predicted monomer model into the cryoET density map, where the interior parallel β-sheets circumferentially extend, and the cap domains repeat every 28 Å (Fig. 1i and Supplementary Movie 2). This SH–SH spacing is identical to that revealed in the TEM image of a sheath component (Fig. 6 in Southam et al. [12]), suggesting the previously isolated sheath component is a β-ring segment. The subtomogram average also reveals that β-rings are assembled into a β-hoop through both β-sheet to β-sheet and linker to β-sheet interactions and that neighboring β-hoops are connected through cap–cap interactions (Fig. 1j). Within a β-hoop, β-sheets interact with neighbors closely while caps do not, consequently, the interior of the β-hoop, formed by the bottoms of the β-sheets, is narrower than the exterior of the β-hoop, formed by the caps; in other words, the axial cross-section of the β-hoop exhibits an annulus sector shape (top panel of Fig. 1j).

### SH contains a motif similar to the β-arch kernel found in amyloid fibrils

The remarkable size of the β-sheets in the *M. hungatei* sheath (Fig. 1i) prompted us to compare the β-ring structures with those in various amyloid cross-β fibrils[19–23]. β-sheets 1 and 2 are separated by 10 Å (Fig. 2a, b), consistent with previous X-ray diffraction and FTIR spectrum observations of archaeal sheaths[14,16,24], which also suggested that the sheath has a high content of structural elements characteristic of amyloid cross-β structure[20,23,25,26]. Recurring in amyloid fibrils is the β-arch motif, in which two β-strands of a continuous polypeptide chain interact via their amino-acid side chains, forming an arch[27,28] that dictates fibril formation, as observed in α-synuclein fibrils[19,21] (Fig. 2c, d). Strands within each β-sheet of the *M. hungatei* SH monomer are arranged in an anti-parallel fashion (Fig. 2b), and β-strands across β-sheets 1 and 2 are connected through β-arches (Fig. 2b). Thus, each SH has 6 β-arches with different side chain compositions, unlike the identical β-arches within a pathological amyloid fibril. In addition, the two β-strands within a β-arch in SH are 15° tilted from each other instead of being parallel as in disease-related β amyloids (Fig. 2c). The SH structure allows side chain interactions between different β-arches instead of forming the typical steric zipper structure in pathological amyloid fibrils. For example, α-synuclein[19] and the Alphafold-predicted

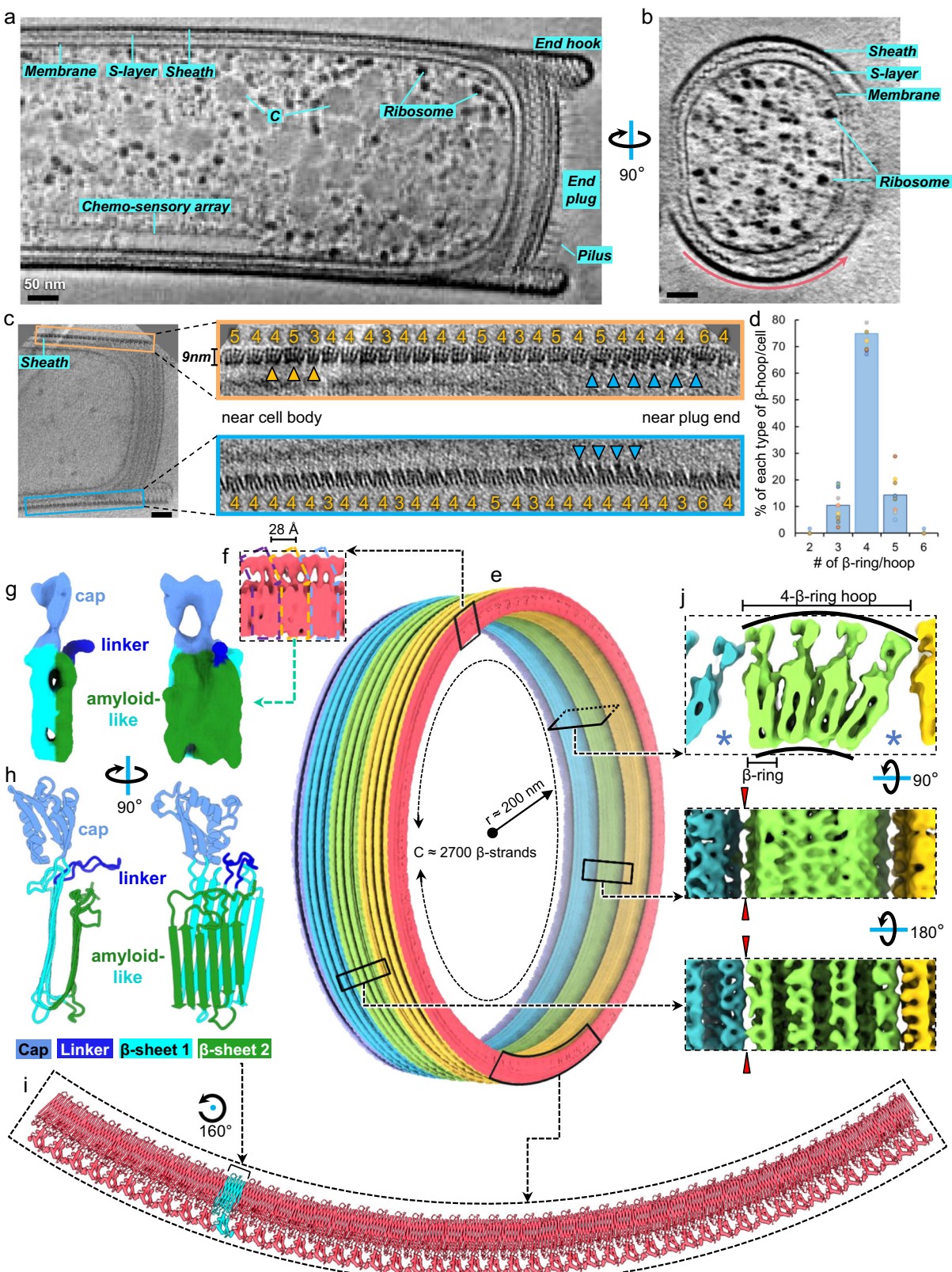

structure of bacterial curli major subunit csgA protein[29]. These unusual β-arch properties indicate that SH can acquire stability through inter-sheet side chain interactions possibly through hydrogen bonds and salt bridges (Fig. 2d and Supplementary Fig. 3), and by forming hydrophobic cores with aromatic amino acids (Fig. 2d).

The cylindrical shape and consistent dimension of the sheath tube raise two significant questions: how could the β-arches in SH form β-rings in archaeal cell sheath instead of β-solenoids in amyloid fibrils, and what determines the radius of the β-ring? Interestingly, at the interface between two adjacent SHs in a β-ring, the two amino acids connecting β-strands S11 to S12 are not part of the β-sheet (Fig. 2e), creating a disruption to β-sheet 2. As a result, instead of participating in backbone interactions as in a typical β-sheet, residues K363 and D43 from adjacent SH subunits could potentially interact through a salt

**Fig. 1 | In situ structure of *M. hungatei* sheath layer. a, b** Two orthogonal cross-sections of a representative cryoET tomogram at lower magnification, showing the overall shape, size, and distribution of the sheath. Intra-cellular structures, such as ribosomes, chemo-sensory array, and membrane-less condensate (C), can be recognized inside the cell. In total, 23 tomograms were collected in this session. **c** Cross-section of a higher magnification tomogram of the region near the plug at one end of a cell. β-hoops (yellow arrowheads) with 3–6 β-rings can be identified in the zoom-in views of the sheath layer and are labeled with 3–6, respectively. Blue arrowheads point to extra densities attached to the β-hoops near the plug. **d** Relative frequency histogram of different types of β-hoops in each cell, overlaid by the corresponding data points. The histogram data are presented as mean values over $n = 10$ tomograms examined in total. β-hoops with 2 or 6 β-rings are very rare

(<1%). **e** Model of a partial sheath layer containing 3 full and 2 half β-hoops colored differently, with circumference (C) equal to the length of around 2700 β-strands and radius (*r*) around 200 nm. The boxes mark the regions and viewing angles of the structures detailed in **f**, **i**, and **j**. **f** Subtomogram average showing the structure that repeats every 28 Å along the outer circumference. **g, h** CryoET density map (**g**) and AlphaFold model (**h**) of the SH monomer. **i** Atomic model of a partial β-ring with one SH monomer highlighted in cyan. **j** Three orthogonal (cross-section, interior, and exterior) views of the subtomogram average of the sheath layer with clefts between adjacent β-hoops and 'holes' (indicated by red arrowheads) at the inter-β-hoop interface. Source data are provided as a Source Data file for (**d**). Scale bar = 50 nm in (**a**–**c**).

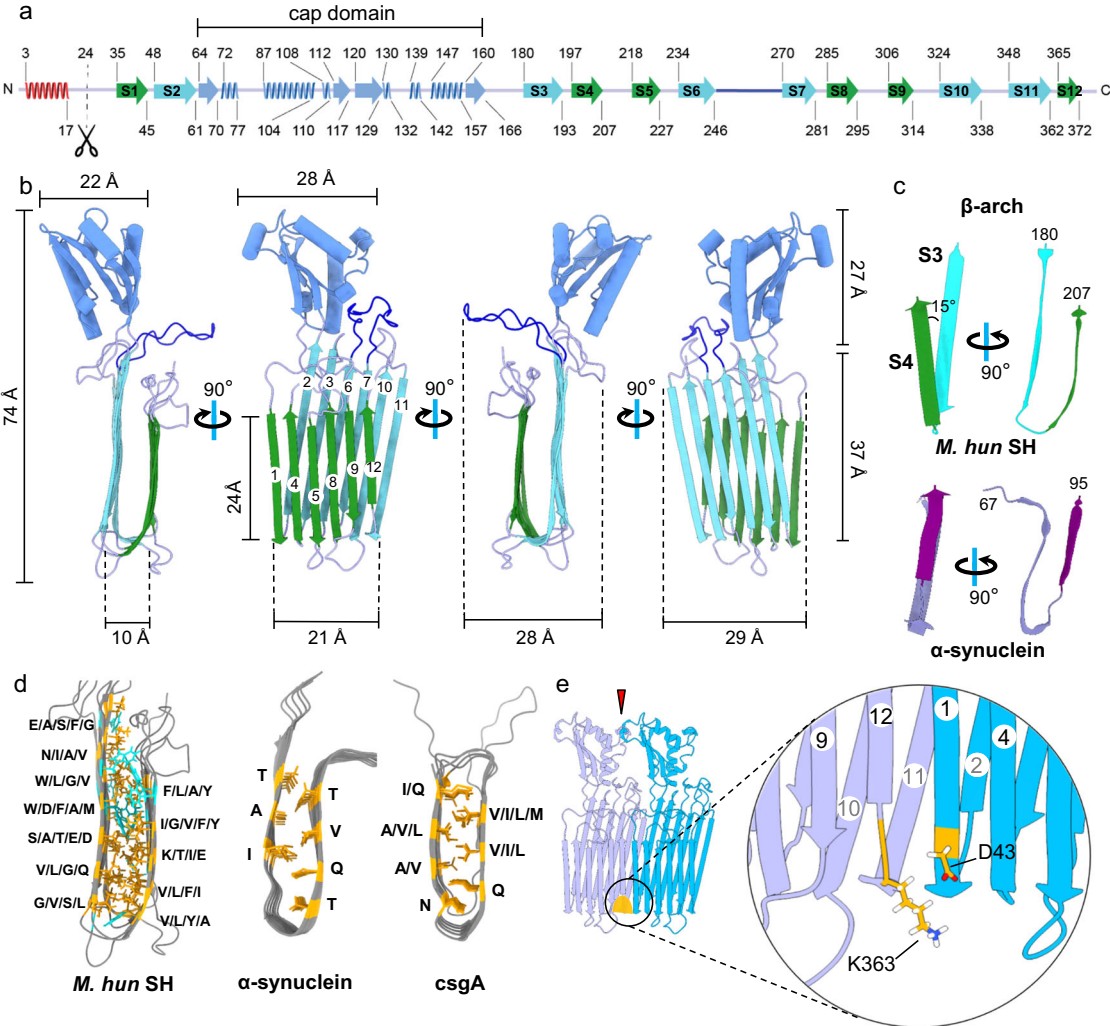

**Fig. 2 | Amyloid properties of SH monomer. a, b** Structure of the SH monomer secondary structures displayed either along its amino-acid sequence (**a**) or as ribbon diagram in 3D (**b**). Different structural elements are colored consistently between (**a**) and (**b**), and the truncated N-terminal signal peptide (red) is shown only in (**a**). **c** Comparison of the β-arch in SH and in α-synuclein (PDB: 7LC9). **d** Comparison of side chains between parallel β-sheets among *M. hungatei* SH, α-

synuclein, and curli major subunit csgA. Aromatic amino acids are colored in cyan and others in orange. **e** Atomic model of two adjacent SH monomers along the outer circumference, with putative cap–cap interaction pointed by arrowhead. Inset shows the disruption on β-sheet 2 (yellow) at the interface between K363 and D43 of the two neighboring monomers.

bridge between their side chains (Fig. 2e inset). Also, cap–cap interactions between adjacent SHs likely cause steric hindrance at the opposite side of this salt bridge between K363 and D43 (Figs. 1i and 2e). These two structural features can lead to a shorter interior rim distance than the exterior rim distance, causing a slight curvature of around 0.8°. The long-range consequence of this curvature is that ~450 SH subunits assemble into a self-limiting circular β-ring (Fig. 1e and

Supplementary Movie 2), in contrast, the α-synuclein forms a non-terminating, continuous amyloid β fibril.

Thus, one β-ring contains a "ring-amyloid" structure comprising ~2700 β-strands for each of its two giant β-sheets, a remarkable β-sheet size only second to that seen in various amyloid cross-β fibrils. Axially, each micrometer of *M. hungatei* sheath structure contains ~89 β-hoops, corresponding to ~356 β-rings or ~160,200 SH subunits.

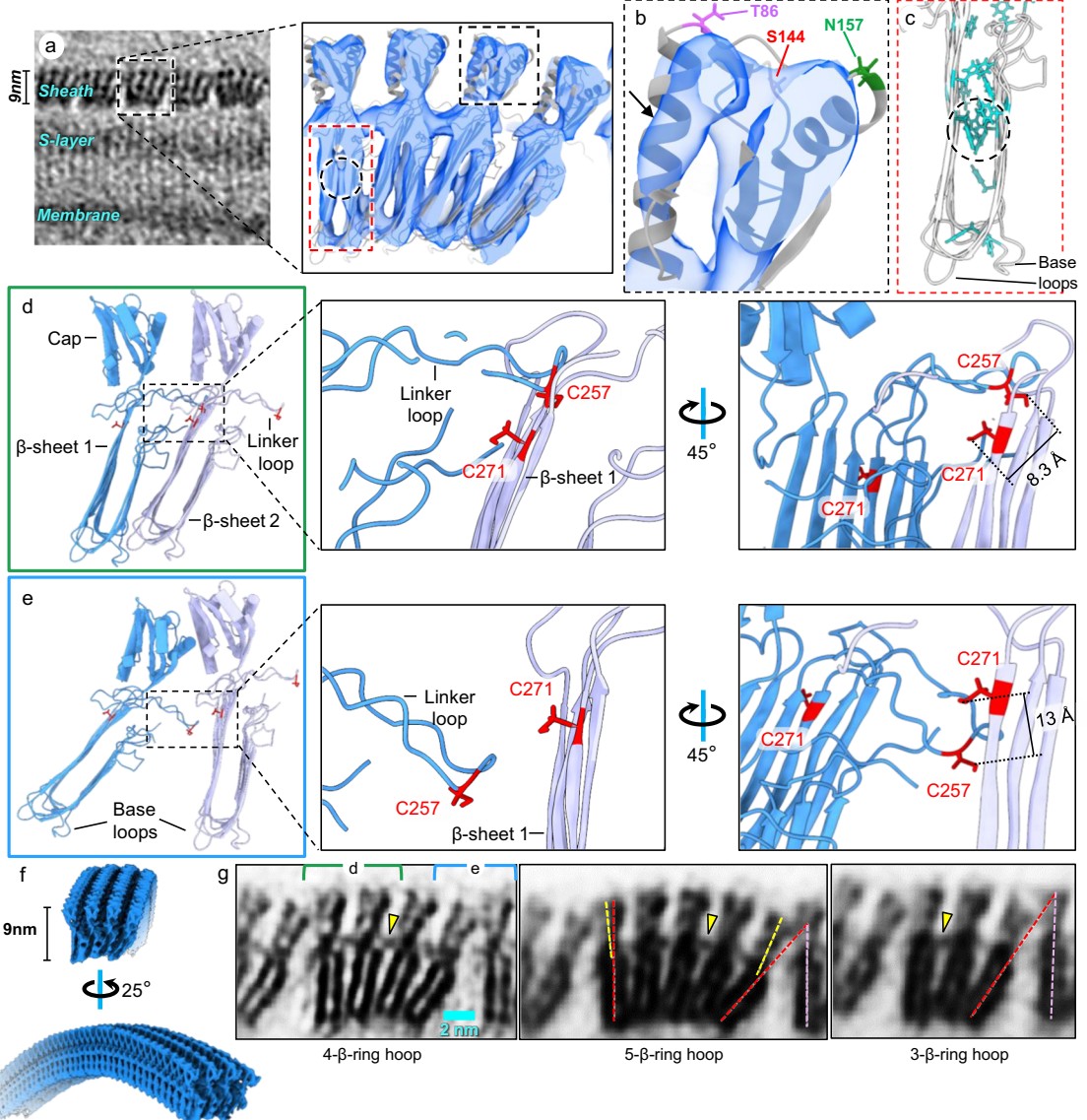

**Fig. 3 | Atomic model of sheath β-hoop. a** Typical distribution of *M. hungatei* cell envelope and zoom-in subtomogram average of the 4-β-ring hoop docked with predicted SH monomer models. The black and red boxes mark the regions of the structures enlarged in panels **b**, **c**, respectively. **b** Enlargement of the cap region in **a**, with its longest helix, measured ~23 Å in length, indicated by an arrow. **c**, All aromatic residues between β-sheet 1 and 2 are colored in cyan, the circled region indicates a cluster of aromatic residues, whose location overlaps with those densities in the subtomogram average that connects two β-sheets in the middle. **d**, **e** Interfaces of two neighboring SH subunits along the cell growth axis either within a β-hoop (**d**) or between two β-hoops (**e**), their locations in the sheath are indicated in (**g**). The insets show the putative disulfide bonds connecting adjacent β-rings. **f** Model of a segment of 4-β-ring hoop, filtered to 10 Å resolution and shown as shaded surface. **g** Subtomogram averages of 4-, 3-, and 5-β-ring hoops, with the intra-hoop SH connection indicated by yellow arrowheads. The angles formed by red and pink dashed lines show the variation near the β-hoop interfaces caused by the number of β-rings in a β-hoop; the angles formed by yellow and red dashed lines suggest a flexible connection between the cap and the β-sheets domain.

Accordingly, the sheath of a 500 µm multi-cellular filament contains about 44,500 β-hoops, 178,000 β-rings, or 80,100,000 SH subunits, representing the largest known protein polymer in a living cell.

## SH interactions within a β-hoop and between adjacent β-hoops

Docking AlphaFold-predicted SH monomer model into the 4-β-ring hoop subtomogram average map reveals how the β-hoops are held together to form a cylindrical sheath layer (Fig. 3a). In the cap domain, a 23 Å-long helix in the AlphaFold-predicted model matches well with a helix resolved in the density map (Fig. 3b); this, together with the matched β-sheets 1 and 2 (Fig. 3a inset), indicates that the atomic model was docked without ambiguity in orientation. Densities were observed between β-sheets 1 and 2, corresponding to a cluster of

aromatic residues between the two β-sheets in the Alphafold model (circled in Fig. 3a inset and Fig. 3c). These extensive connections between β-sheets 1 and 2, together with potential salt-bridges between charge pairs (aspartic acid-lysine and glutamic acid-lysine) (Supplementary Fig. 3), stabilize the amyloid-like β-ring structure. The docking also reveals possible sites of glycosylation[4], where amino acids Thr86, Ser144, and Asn157 in the cap domain are exposed to the exterior of the cell (Fig. 3b).

The β-ring arrangement in a β-hoop suggests the base loops (Fig. 3c) in the β-sheet domain may serve multiple functions. First, the two extending loops at the left and right bottom of the SH monomer could form extra connections and establish the binding point when β-rings start assembling into a β-hoop (Fig. 3c and Supplementary

Fig. 4a, b). Second, the asymmetric arrangement leaves a free-charged loop on both the left and right sides of a β-hoop (Supplementary Figs. 4a, b). Coincidentally, extra densities attached to the sheath interior at the same place are only observed near the plug region (blue arrowheads in Fig. 1c insets), suggesting that the exposed base loops could serve as binding sites for proteins that regulate plug formation and/or cell division.

The aforementioned linker connections between adjacent β-rings were analyzed by examining the spatial relationship between two adjacent SH monomers to localize the previously speculated disulfide bond[11,15,30] (Fig. 3d, e). To illustrate the shape, organization, and inter-SH interactions, we built a 4-β-ring hoop segment (Fig. 3f, Supplementary Figs. 4a, 5 and Supplementary Movie 3) by placing the above β-ring segment models into the tomogram and following the β-ring stacking arrangement within a β-hoop visible in the TEM image of flattened sheath layer[24]. The model of this hoop segment reveals that each SH monomer contains two cysteine residues[7] on opposite sides, one on β-sheet 1 and the other on the linker. These cysteine residues from neighboring SH subunits both within a β-hoop (Fig. 3d) and between two β-hoops (Fig. 3e) are only 8.3 and 13 Å apart, respectively. Although the distances between β-rings inside a β-hoop (i.e., intra-hoop β-rings) are longer than that of a typical disulfide bond, Cys257 is at the tip of the linker loop, whose flexibility should allow the formation of a disulfide bond between the two cysteine residues. When two β-ring models are aligned by this putative disulfide bond, the complementary electrostatic potential distribution at their interfaces further supports the above β-hoop assembly pattern (Supplementary Fig. 5b). Indeed, there is clear density in the linker region between intra-hoop β-rings (arrowhead in Fig. 3g) but such density is missing between inter-hoop β-rings. Thus, we propose that disulfide bonds connect SH monomers between intra-hoop β-rings, providing strong interactions to impart structural rigidity for the β-hoop.

Beyond rigidity, the existence of different β-hoop types raises another vital question: how the sheath establish axial linearity leading to the cylindrical shape of *M. hungatei* cells (Fig. 1c inset and 3a)? The variation of numbers of β-ring per β-hoop results in different sizes of the axial cross-sectional annulus sectors among these β-hoops, e.g., a 5-β-ring hoop has a longer annulus sector arc than a 3-β-ring or 4-β-ring hoop (Fig. 3g). These different sizes would seem to create varied contact points between different β-hoop types, which could affect the molecular bonds between β-hoops. Therefore, we compared the subtomogram averages of 4-β-ring, 5-β-ring, and 3-β-ring hoop to investigate the local subunit conformational changes and inter-β-hoop interfaces (Fig. 3g).

Adjacent β-rings contact each other not only at the base of the β-sheet domain but also in the middle through the linker, probably involving disulfide bonds (Fig. 3d, g). As shown in Fig. 3g, the only contact point between neighboring β-hoops is at the cap domains of their outermost β-rings and remains the same; but the angle formed between them (i.e., the angle formed by red and pink dashed lines in Fig. 3g) is larger for β-hoops containing more β-rings.

Within each β-hoop, the first β-ring (the leftmost of the 4-β-ring hoop in Fig. 3g) is perpendicular to the axis of the cylinder, and the subsequent β-rings are increasingly tilted to the right while the cap domains within these β-rings are increasingly less tilted as compared to their β-sheets domains (see the relationship between the yellow and red dashed line in Fig. 3g middle panel), suggesting a flexible connection between the cap and the β-sheets domain. Such flexibility would compensate for differences in the angles between adjacent β-hoops to maintain consistent molecular contact between contacting SH subunits, and also allow all β-hoops to align linearly along the axial direction of the cylindrical cell.

### In situ structure of an immature cell

By imaging actively dividing *M. hungatei* cells, we sought to understand how newly translated SH proteins are transported across the membrane and protein layers to assemble into such well-organized polymers outside the cells. The *M. hungatei* replicates through dividing a cell into two daughter cells such that the sheath layer encapsulating multiple cells elongates concomitantly with the cell growth. From one of our tomograms (Supplementary Movie 4), we observed an immature cell measured around 480 nm long (Fig. 4a), much shorter than that of the mature cell (Fig. 1a). Its nascent plug component is tilted and not fully sealed near the sheath layer (Fig. 4a), unlike that in the mature cell[3,5,12]. Further, between the S-layer and the cell membrane is a pocket that is absent in the mature cell (Fig. 4a); the space within the pocket harbors densities which, as detailed below, we interpret as nascent SH polymers assembled immediately after SH synthesis and membrane translocation. These features are consistent with an immature cell.

To better observe the organization of this immature cell, a 3D rendering of the tomogram was generated (Supplementary Movie 5). Several filamentous densities with similar thicknesses around 10 nm were observed both within the pocket, adjoining the cell membrane and the S-layer (Fig. 4b, c), and in the interstitial space between the plugs of the two adjacent cells (Fig. 4d). Within the pocket, the filamentous densities have several contact points with the cell membrane, and the S-layer appears to be discontinuous near the pocket and there is a gap between the newly forming plug layers and the sheath (Fig. 4b, c); in the interstitial space, the filamentous densities have a curvature that is similar to the curvature of the β-hoops, though these densities are not as aligned as the β-hoops in the sheath (Fig. 4e). Additionally, close to these densities are unfilled lens-like openings between adjacent sheath layers, where β-hoops are partially separated (Fig. 4f); and such an opening has never been observed in normal mature cells before. Thus, we propose that those filamentous densities are the precursors of β-rings in different stages: the pocket ones, given their proximity to the membrane, are likely newly synthesized SH polymers; and the interstitial ones, given their location next to the opening in the sheath layer, are partially aligned β-rings in the processing of assembling into β-hoops that could be subsequently inserted into the opening, driving sheath elongation and cell growth. In the absence of direct evidence establishing the identity of these densities, it is also possible that these filamentous densities are other extracellular polymers, such as flagella, with a diameter of around 10 nm.

## Discussion

By providing a barrier against entropy flow as governed by the second law of thermodynamics, the compartmentalization of biological molecules and functions is synonymous with life's emergence on Earth. Membrane targeting through ribosome-bound Sec61/SecYEG complex enables delivery of trans-membrane proteins to serve as conduits across the membrane, typical of the situation for eukaryotic cells[31]. In bacteria, some of these trans-membrane proteins act as cell-wall synthesis enzymes, and others as lipid transporters for the assembly of the outer membrane of Gram-negative bacterial cells. Highly ordered protein polymers typically are assembled at the location of protein synthesis within the confines of the cytoplasm and regulated by various protein or non-protein factors when a dynamic process between assembly and disassembly is necessary. A prominent example of a highly ordered protein assembly is the protein capsid of non-enveloped viruses, which typically manifest mostly as spherical assemblies with icosahedral symmetry[32–37] and occasionally as filamentous assemblies with helical symmetry[38]. Such complexes assemble within their host cytoplasmic site immediately after protein translation and subsequently translocate across the cell membrane as fully assembled virions either by exocytosis of cells without cell walls or by lysis of cells from within cell walls (as in bacteria and eukaryotic plant cells). Distinctive from eukaryotic and bacterial cells, the existence of external, highly ordered proteinaceous layers in archaeal cells poses additional challenges to ensure

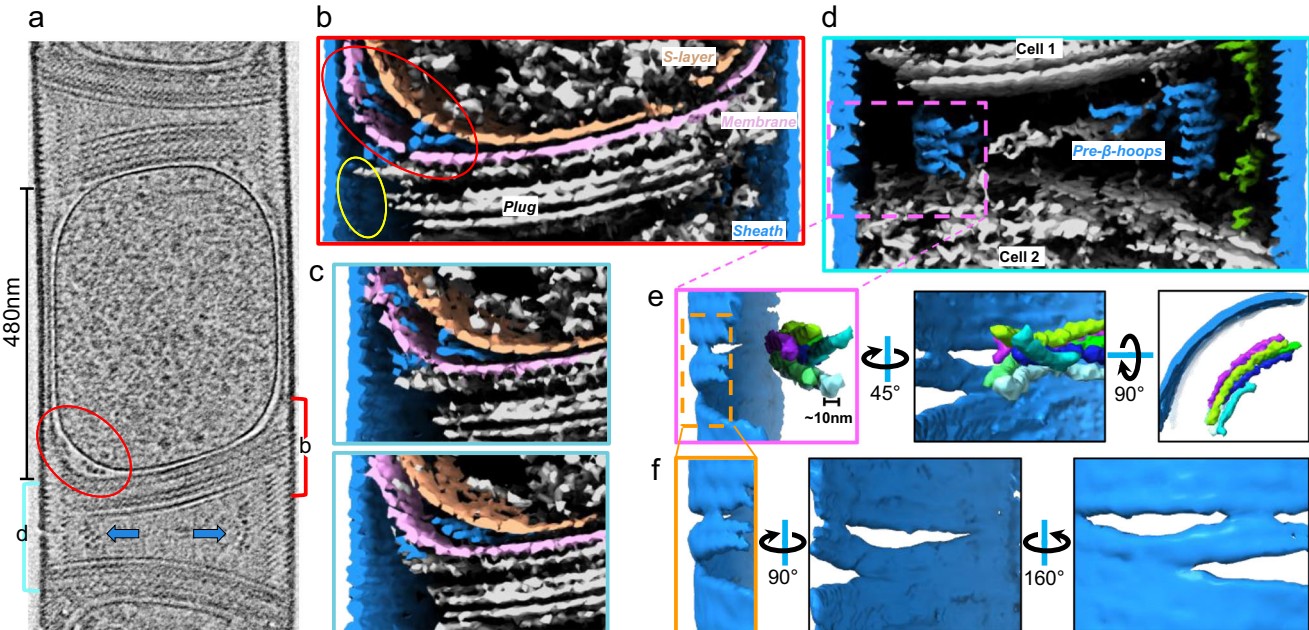

**Fig. 4 | In situ structure of an immature cell. a** Tomogram of an immature *M. hungatei* cell with a cell length around 480 nm, tilted incomplete plugs, and pocket area (red ellipse) located between the membrane and S-layer. Filamentous densities (indicated by blue arrows) both appear inside the pocket area and between plugs from neighboring cells. The upper and lower boundaries of the segmented structure displayed in **b** and **d** are indicated by red and cyan brackets, respectively. In total, 37 tomograms were collected in this session. **b–f** Cross-section views at different z of 3D-rendered map generated from the tomogram in **a**, showing filamentous or putative pre-β-hoop densities (color in blue) either inside the pocket near the plug breakage (yellow ellipse) on the left (**b**, **c**), or in the interstitial space between plugs (**d**). Putative pre-β-hoop densities with similar curvature as the outer sheath are colored individually (**e**), and their location is close to the temporary sheath opening region (**f**).

ordered assembly and timely delivery of such structures from the inside of the cell to the outside.

Our study shows that the outmost layer of *M. hungatei* cell is composed of a single small protein of just 40 kDa with an amyloid-like cross-β structure, thus forming a functional amyloid polymer. Other microbial functional amyloids are known, including the extracellular fibrillar structures described in both Gram-negative and positive bacteria with roles in biofilm formation and adhesion, for example, the curli, Bap and Esp orthologs[39], P1 adhesions, Harpins and modulins (see review[40]). Many structural and functional aspects of their biological roles remain to be fully elucidated. How does the *M. hungatei* SH amyloid-like molecule assemble into a biologically functional cylinder? The presence of additional loops at the base of its β-arch plus the "linker" loop, both absent in pathogenic amyloids, appear to direct the assembly of the β-ring and β-hoop. The subsequent axial stacking of many multi-ring β-hoops, each of which contains thousands of amyloid-like β-arches yields in essence, a gigantic supramolecular structure composed of ~13,456,800 β-strands for a typical mature *M. hungatei* cell of ~7 μm in length. Axially, each micrometer of the cylinder contains ~89 β-hoops, corresponding to ~356 β-rings or ~160,200 SH subunits. Thus, a 500 μm multi-celled filament contains about 44,500 β-hoops, 178,000 β-ring. While both are evolutionarily successful, the use of a single small protein to encase and protect the cell in *M. hungatei* is a genetically simpler solution than bacterial peptidoglycan-containing cell envelope, which requires numerous genes, complex synthesis-assembly pathway, and associated regulatory system to accomplish.

Though much remains to be clarified and the temporary opening in the sheath needs to be verified, the visualization of the "growing" sheath in dividing cells provides some clues about the archaeal sheath assembly. One provisional model is a four-stage model of sheath biogenesis as illustrated in Fig. 5. SH monomers are synthesized by polyribosomes near the inner surface of the cell membrane and SH signal peptide (Fig. 2a) at the N-terminus directs secretion of these monomers through a Sec61/SecYEG translocon[41] or a secretion system encoded by one of the four sets of type II/IV secretion genes within the *M. hungatei* genome[7]. Cleavage of the signal peptide triggers SH folding into a cross-β rich structure, which oligomerizes into short β-ring segments through β-sheet augmentation (Fig. 2e). These short β-ring segments could pass through the S-layer through the pocket region, then elongate near the incomplete nascent plug. Upon arriving at the interstitial space between two daughter cells, the newly assembled short β-ring segments stack to form pre-β-hoops and insert into a temporary opening in the sheath layer between two cells, elongating the sheath to provide more space for cell growth (Fig. 5a, b). These steps are based on the observations in tomograms of immature cells (Fig. 4, Supplementary Movies 4 and 6). Notably, our model of sheath synthesis, assembly, and translocation is consistent with the distinctive property of many archaeal viruses—they fashion a highly ordered proteinaceous capsid (likely derived from archaeal cells[42]) external to a bilayer membrane without transmembrane viral proteins. Nonetheless, due to the intrinsic lack of temporal progression information in the frozen-hydrated sample used in cryoET, this proposed sheath biogenesis should only be considered as a putative model to be tested by orthogonal methods such as time-resolved fluorescence light microscopy.

As contemplated by quantum physicist Erwin Schrodinger in his 1944 book entitled "What is Life?", the crystalline arrangement of simple molecules provides the means to overcome increasing entropy in order for life to emerge from chaos. We now know that perfect crystalline arrangement, including three-dimensional forms as in table salt and in one-dimensional form as in amyloid β-fibrils, is not a solution to the many functional and dynamic processes of living organisms. Rather, some minor deviation from perfection might just be what is necessary to drive these processes. Archaea are considered extremophiles that are related to Earth's primitive atmosphere and typically exist in extreme habitats. Found in sewage sludge and as a model organism of the Archaea, *M. hungatei* has been used for examining

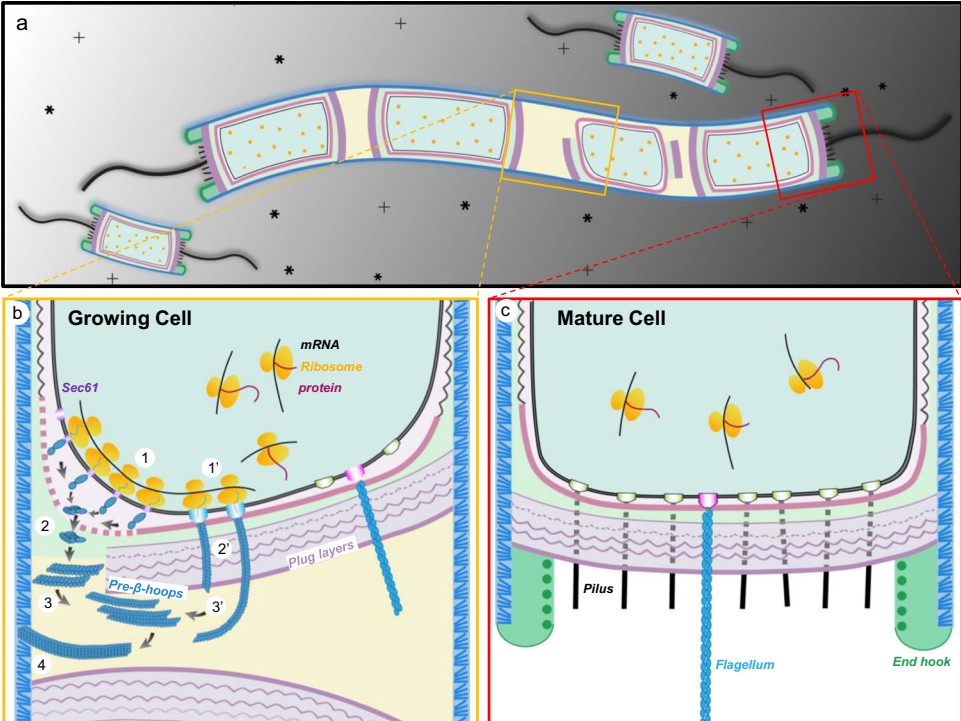

**Fig. 5 | Illustration of proposed sheath biogenesis of *M. hungatei* cell.**
**a** Schematic illustration of the living environment of *M. hungatei*. The cell filament in the middle contains 1 immature and 3 mature cells. **b** Proposed model of sheath biogenesis depicting four stages: **1**. SH monomer production and pre-β-hoop oligomerization; **2**. pre-β-hoop translocation through either the S-layer breakage or secretion systems on the plug into the pocket and interstitial spaces between plugs; **3**. pre-β-hoop assembly into β-hoops; and **4**. β-hoop insertion into the sheath layer. **c** Organization of the plug region of a mature end. An alternative interpretation of the filamentous densities is the archaeal flagella, as indicated in **b** and **c**, with approximately the same diameter of a 4-β-ring hoop (~10 nm).

syntrophic symbiosis which drives anaerobic carbon recycling in nature. Our results might offer insight into how *M. hungatei* polymerizes the small SH into the sheath with small deviations from crystallinity to accomplish functional tasks. *M. hungatei* sheath has the extraordinary capability of resisting ~300 atmospheric pressure[17], yet should allow the exchange of methane, hydrogen, and water to support basic intracellular functions. Notably, the trunk of buoyancy-control gas vesicles within bacterial and archaeal cells is also a thin-walled cylinder with distinctive hydrophilic exterior and hydrophobic interior assembled from a single β-hairpin-containing protein[43,44], thus differs from the archaeal sheath which fashions alternating hydrophobic/hydrophilic patches (Supplementary Fig. 6). In archaeal sheath, the intra-hoop SH interactions are extensive, explaining its capability to resist high pressure. The interior of each β-hoop has no holes and thus is well sealed by its amyloid-like β-sheet domains (Fig. 1e and j), therefore the wedge-shaped cleft between β-hoops, which varies in shape and measures up to ~30° in angle and 40 Å in width (Fig. 1j), appears to be the only outlet for waste gas. Many questions still remain concerning the details of biogenesis, biophysical properties, and regulatory mechanisms of the sheath. Nonetheless, from the grandeur perspective, SH's deviation from a "perfect" β-sheet (Fig. 2e) and crystalline packing contributes to the cylindrical shape of the sheath, thus its multiple functional roles: determining the cell size and cylindrical shape, serving as an environmental barrier against harmful agents, creating a novel periplasmic space bounded by the SH and S-layer, and providing a rigid cell platform for taxis-directed flagellar movement. Thus, while Mendel and Morgan's discoveries of genetics provided the answer for variation missing in Darwin's theory to drive the evolution of life, the current work illustrates an example of how a small 40 kDa protein introduces variation to perfect periodicity/crystallinity to impart functions of life processes.

## Methods

### *M. hungatei* cell growth and sample preparation

*M. hungatei* strain JF1 (ATCC 27890) was cultured anaerobically in 28 ml anaerobic tubes with a vessel headspace pressurized to 10 psi with an 80:20 (vol/vol) mixture of $H_2:CO_2$[4]. Each 1000 ml of medium contained the following: 0.54 g NaCl, 0.12 g $MgSO_4 \cdot 7H_2O$, 5.0 g $NH_4Cl$, 1.8 g $KH_2PO_4$, 2.9 g $K_2HPO_4$, 0.06 g $CaCl_2 \cdot 2H_2O$, 2.72 g Na Acetate·$3H_2O$, 10 mL of 100× trace metal solution and 1 ml of 1000× vitamin solution. Following sterilization, the medium was supplemented with a 10 ml filter-sterilized solution of reducing reagent (2.5% $Na_2S \cdot 9H_2O$, 2.5% Cysteine HCl) and 20 ml of a 1 M $NaHCO_3$ solution. Following inoculation, anaerobic tubes containing 10 ml of medium were incubated at 37 °C horizontally on a rotary drum shaker (60 RPM, New Brunswick, Inc). Cells were serially transferred at least three times, with transfers made at the mid-exponential phase, to achieve 10+ cell doublings prior to harvest. To capture dividing cells, cells were harvested early in the exponential growth phase by centrifuging 1 ml of cell suspension at low speed (5000×*g*) and resuspending in 100 µl of medium over a 3-day period, with 24 h between each harvest.

### CryoET and tomogram reconstruction

*M. hungatei* samples were mixed with 5 nm-diameter fiducial gold beads. 3 µl of the mixture was applied onto Quantifoil (3:1) holey carbon grids that were freshly glow-discharged for 30 s at −40 mA. With an FEI Mark IV Vitrobot cryo-sample plunger, the excess sample on the grid was blotted away with filter paper at a blot force of −4 and blot time of 5 s. The sample was vitrified immediately by being plunged into liquid nitrogen-cooled liquid ethane. Plunge-freezing conditions and cell concentration on the grids were optimized with an FEI T20 transmission electron microscope equipped with an Eagle

**Table 1 | CryoET data collection and processing statistics**

|  | High mag. (whole cell) | Low mag. (whole cell) | Low mag. (immature cell) |
|---|---|---|---|
| *Data collection* | | | |
| Microscope | Titan Krios | Titan Krios | Titan Krios |
| Voltage (kV) | 300 | 300 | 300 |
| Volta phase plate (VPP)[54] | No | Yes | No |
| Total electron exposure ($e^-/Å^2$) | 110 | 110 | 100 |
| Slit width (eV) | 20 | 20 | 20 |
| Detector | K2 | K2 | 4 mega pixel CCD |
| Defocus range (μm) | −1.5 to −4.0 | N.D. | −3.0 to −6.0 |
| Pixel size (Å) | 1.634 | 4.050 | 6.102 |
| Software | SerialEM[45] | SerialEM[45] | Batch Tomography |
| Tilt-series range | ±60° | ±60° | ±70° |
| Tilt-series increment | ±3° | ±3° | ±2° |
| Tilt-series scheme | Bi-directional | Bi-directional | Continuous |
| Tilt-series used | 10 | 1/23 | 1/37 |
| *Data processing* | | | |
| Software: tilt-series alignment | IMOD[47] | IMOD[47] | IMOD[47] |
| Software: final reconstruction | Relion4.0[50] | N.D. | N.D. |
| Initial particle images: 3, 4, 5-β-ring hoop (no.) | 3,323, 20,137, and 4,237 | N.D. | N.D. |
| Final particle images: 3, 4, 5-β-ring hoop (no.) | 2,971, 17,359, and 3,609 | N.D. | N.D. |
| Final Box-size (px) | 120, 160, 120 | N.D. | N.D. |
| Pixel size final reconstruction (Å) | 3.268, 1.634, 3.268 | N.D. | N.D. |
| Symmetry imposed | C1 | N.D. | N.D. |
| Map resolution: 3, 4, 5-β-ring hoop (Å) | 21.8, 7.9, and 14.0 | N.D. | N.D. |
| FSC threshold | 0.143 | N.D. | N.D. |

N.D.: not determined

2K HS CCD camera. Grids with vitrified cells were stored in a liquid nitrogen dewar until use.

With either FEI Batch Tomography (for lower Mag. Data collection) or *SerialEM*[45] (for higher Mag. Data collection), tilt series were collected in a Titan Krios instrument equipped with a Gatan imaging filter (GIF) and a post-GIF K2 direct electron detector in electron-counting mode at California NanoSystems Institute (CNSI); the data collection parameters are listed in Table 1. Frames in each movie of the raw tilt series were aligned, drift-corrected, and averaged with *Motioncor2*[46]. The tilt series micrographs were aligned and reconstructed into 3D tomograms using the *IMOD* software package[47].

### Subtomogram averaging
Subtomogram averaging was performed with *PEET*[48,49] and *Relion4*[50]. In situ details of the sheath layer and its assembly pattern along the longitudinal axis were visible in our tomograms. However, because of the preferred orientation of the *M. hungatei* cells (Supplementary Fig. 1b) and the missing wedge effect, the SH monomer repeat unit along the outer circumference was not distinguishable. Learned from both the previous study that described the diffraction patterns of the isolated β-hoops[14], and the dimension of the AlphaFold-predicted model, the SH monomer repeat unit along the outer circumference was presumed to be 3 nm for the later particle picking process.

Initial particles, i.e., sub-tomograms, of the 3-β-ring, 4-β-ring and 5-β-ring hoops were manually picked, one per β-hoop, at the z-coordinate where they were most visible. The picked particles were averaged with *PEET*; 68, 465, and 85 particles were used to generate the initial reference maps for the 3-β-ring, 4-β-ring, and 5-β-ring hoop, respectively. The geometry shown in the initial reference map, combined with the proposed 3 nm repeat unit of SH monomers, guided us to sample more particles circumferentially along the β-hoops to expand the particle set. *PEET* was used to align the final expanded particle set, resulting in one *PEET* motive list file per tomogram (i.e., *MOTL.csv file), containing translational and angular alignment information.

Then, we first use *createAlignedModel* to generate new *PEET* motive list files to re-center all particles and later use *MOTL2Relion* to convert the above angular alignment results stored in the motive list file into *Relion's* convention. Such that the coordinates and orientations of the particles were formatted and imported into *Relion4* for further refinement. Following one round of 3D refine under 4-binned pixel size and two rounds of 3D refine under 2-binned or original pixel size, along with the removal of particles within 2.5 nm from each other to prevent particle duplication, the resulting resolutions for the 3-β-ring, 4-β-ring and 5-β-ring hoop were 21.8, 7.9 and 14.0 Å (Supplementary Fig. 1c–e), respectively. Resolution was calculated on 3DFSC Processing Server[51] and the global resolution reported above is based on the "gold standard" refinement procedures and the 0.143 Fourier shell correlation (FSC) criterion.

### Modeling and 3D visualization
The AlphaFold models for the SH monomer (Fig. 1h) and csgA monomer (Fig. 2d) were generated with AlphaFold2 Google Colab[18]. Tomograms displayed in Figs. 1a, b and 4a were missing wedge corrected by *IsoNet*[52]. Visualization of tomograms and averaged electron density maps were done with *IMOD* and *UCSF ChimeraX*[53], respectively.

### Reporting summary
Further information on research design is available in the Nature Portfolio Reporting Summary linked to this article.

## Data availability
The subtomogram average structure data generated during the current study have been deposited in the Electron Microscopy Data Bank (EMDB) repository, with the accession codes EMD-29442 (4-β-ring hoop), EMD-29443 (3-β-ring hoop), and EMD-29448 (5-β-ring hoop). The previously published structure of α-synuclein shown in Fig. 2d is available in the Protein Data Bank (PDB) repository under accession code 7LC9. The major curlin subunit predicted atomic model shown in Fig. 2d is available in AlphaFoldDB with accession code AF-P28307-F1. The atomic model of *Bacillus megaterium* gas vesicle segment shown in Supplementary Fig. 6 is available in PDB with accession code 7R1C. Source data are provided with this paper.

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

## Acknowledgements

We thank Zhu Si for initial structural determination efforts and Titania Nguyen for editorial assistance. This project is supported by grants from the US National Institutes of Health (NIH) (GM071940 to Z.H.Z.), US National Science Foundation (1515843 and 1911781 to R.G.), and U.S. Department of Energy (DOE) Office of Science (BER) (contract DE-FC-02-02ER63421 to R.G.). We acknowledge use of resources in the Electron Imaging Center for Nanomachines supported by UCLA and grants from the NIH (1S10OD018111 to Z.H.Z.) and the National Science Foundation (DBI-1338135 and DMR-1548924 to Z.H.Z.).

## Author contributions

R.G. and Z.H.Z. initialized and supervised research; F.S., J.Z., H.W., and R.G. prepared samples; D.T., J.Z., and H.W. carried out cryoET imaging and structure determination; S.L. helped data processing and figure preparation; Z.H.Z., R.G., H.W, J.Z., and S.L. interpreted the data and wrote the manuscript; all authors reviewed and approved the paper.

## Competing interests

The authors declare no competing interests.
