## [Peer Review File · Nature Communications]

REVIEWER COMMENTS

Reviewer #1 (Remarks to the Author):

Wang et al report a beautiful in situ structural characterisation of an archaeobacterial sheath protein with a modest number of tomograms (~12 tomograms). Sutomogram averaging to 7.9 Å resolution enabled them to resolve the domain organisation of the sheath (cap, beta-sheet domain, and linker). The authors then use an alphafold predicted atomic model to perform additional interpretation of the organisation of the sheath protein within their in-situ tomographic maps.

In one of the tomographic volumes, a smaller cell is observed that contains openings in the sheath near to the cell, an incomplete plug and some protein filaments. The authors speculate that these are intermediates of sheath assembly.

While the tomographic data is of a high quality, I am concerned that a several of the main points in the manuscript do not have sufficient evidence to support their conclusions:

1) Hydrogen bonds in the amyloid core of the protein (Figure 3).

The authors interrogate an alphafold fold model of SH and describe a hydrogen bonding network linking the two beta-sheets of the amyloid core of SH. This seems to be highly speculative, particularly given the known limitations of alphafold to predict such fine details of a protein. The authors must at the very least present evidence that their model does not contain atomic clashes and be more cautious in their description of this feature in the alphafold model.

2) Disulphide bonds in the sheath (Figure 3).

Using the alphafold model of SH and placing this in their subtomogram map, the authors note that this places cysteine residues from neighbouring SH molecules 8-13 Å apart. They then assert that the linker is flexible and has formed a disulphide bond. Surely if the linker region had formed a disulphide bond, then it would no longer be as flexible and tomographic density from the one SH protein to its neighbour would be well-resolved? I wonder if the authors might consider providing biochemical evidence to support this assertion by blocking free thiols with iodoacetamide, then running an SDS-PAGE gel without a reducing agent and comparing it to an SDS-PAGE gel with a reducing agent. Presumably if the SH protein formed intermolecular disulphides, you would not see a monomer band (by immunoblot) in the absence of a reducing agent.

3) Putative intermediate Pre beta-hoop (Figure 4).

The authors discuss a single tomogram that includes a smaller cell than the others in their 10 tomogram dataset. In it they conclude that this is a 'growing cell' and identify filamentous material between the cell membrane and sheath that they describe as 'Pre-beta-loops'. This seems highly speculative. How can they be sure that this rod-shaped tomographic density is formed of SH and not some other extracellular protein? This must be much more cautiously stated. Perhaps this could be supported with more convincing evidence.

Overall, the manuscript was also very difficult to follow, with numerous sentences that contain errors or were ambiguous/impossible to interpret. Some of the figures are also unclear with incomplete figure legends. I wonder if the first draft of the manuscript was written by the first author (there is certainly

considerable enthusiasm, which I applaud!) but perhaps the lead author, who is highly experienced and a world leader in cryoEM, has not given this manuscript sufficient attention to getting it in a form that is ready for peer review.

Below are some specific examples (but by no means an exhaustive list) of errors/problems in the manuscript (I have highlighted the more significant problems with a ***):

In the abstract

Lines 2-4: The first sentence is unclear.

Line 13: What does “azimuthally augmented” mean?

***Lines 14-18: “Tomograms of dividing cells...”. This sentence misrepresents the data in the manuscript. None of the tomograms showed evidence of a ‘dividing cell’. Only one tomogram showed a smaller cell with holes in the sheath and filamentous material in the space between two plugs. No structural data showing ‘insertion’ of beta-hoops’ was presented in the manuscript.

In the introduction:

Line 21: “cells in Eukarya have a simple lipid membrane”. This is an over-simplification. What about plant cells?

Line 32: “500 um”. Is this dimension referring to length?

Line 37: “Its cylindrical shape is essential for optimizing gas exchange...”. No citation of the literature is given to backup this statement. This point also seems tangential to the results presented in the manuscript.

Line 51: “Indeed, the seemingly unrelated characteristics...”. I did not understand this sentence. In particular, what are the unrelated characteristics?

Line 56: “What selective advantage does such an organization provide for cell growth and gas regulation?”. There are no experiments reported in this manuscript that address this question.

Line 60: “oligomerize azimuthally around the cylindrical cell into a ring structure”. This sentence is unclear.

Line 61: What is a “ β -arch kernel”?

In the Results:

***Line 84: “explaining the failure of the early effort to resolve the sheath structure based on crystallinity.” I don’t think one can be so definitive about the limitations of another groups work.

Line 88: “No long-range periodicity in the arrangement of different membered β -hoops is observed along the axial direction (Fig. 1c insets)”. What is long range periodicity? Is this referring to the distribution of different sized β -hoops and them not being arranged with a partern? If yes, I don’t think the information contained fig. 1c properly/thoroughly demonstrates this point.

Line 90: “sorted out”. What does mean?

Line 96: “Repeat units appear”. What are the repeat units?

Line 107: “reveals”. The tense in much of manuscript jumps about from the past to the present tense. Please be consistent.

Line 108: “ β rings alternates along the axial axis”. What do the authors mean by ‘alternates’?

Line 113: “arch shape (top panel of Fig. 1j)”. I could not identify an arch shape in this figure. Please explain or improve this figure.

Line 129: “SH can acquire more self-stability”. More self-stability than what? I am also a little unclear what is self-stability.

Line 131: “form more hydrophobic cores with participation of a large amount of aromatic amino acids”. More hydrophobic cores than what? Please explain. Also, what is a large amount of hydrophobic residues?

Line 136: “is two residues shorter at the fold area, which leaves notch”. I don’t understand this statement. I am also not sure that the word ‘notch’ makes much sense here. Line 137: “ β -strands forming a β - sheet normally only connect via the polypeptide backbone interactions”. I don’t understand what ‘connect’ means.

Line 140: “Also, cap-cap interactions likely reinforce this K136-D43 interaction”. It does not make biophysical sense to describe the distal cap-cap interaction as ‘reinforcing’ the K136-D43 interaction.

Line 154: “In the cap domain, a 23Å-long helix in the AlphaFold-predicted model matches well...(Fig. 3b)”. In fig 3b, is this the helix described as the ‘longest’ in the figure legend or one of the other helices in the cap domain?

***Line 157: “corresponding to the predicted locations of hydrogen bond core between the side chains of amino acids near the center (Fig. 3a red box and Fig. 3c).” This sentence is poorly written. What is a ‘hydrogen bond core’. Fig. 3c needs to be improved: the presence of hydrogen bonds is not clear. I suspect this might be over-interpretation of an alphafold model since usually the precision of Alphafold is not sufficient to reliably interrogate side-chain-side chain interactions. Have you performed an energy minimisation of the model and ensured that there are no steric clashes of sidechain atoms?

Line 173: “locational relationship”. What does this mean?

Line 176: “and following axial arrangement of SH”. This is unclear.

Line 194: “To answer this question, we compare the subtomogram 95 averages of 4-β-ring, 5-β-ring, and 3-β-ring hoop to investigate the local subunit conformational changes and inter-β-hoop interfaces (Fig. 3g)”. What do the ‘star’ symbols and dashed lines represent in Fig 3g?

Line 197: “connected by β-sheets domain”. This is unclear.

Line 198: “between adjacent β-hoops, the contact points at the caps between neighboring β-hoops seem to remain the same”. The same as what?

Line 202: “Such flexibility would compensate for differences in the angles between adjacent β-hoops to maintain consistent molecular bonds between contacting caps, and allow all β-hoops to be aligned along the axial direction”. What is the evidence that there are consistent molecular bonds between contacting caps?

Line 212: “we observed a growing cell measured around 480 nm long (Fig. 4a), much shorter than that of the mature cell (Fig. 1a)”. You have not presented any evidence that the cell is growing. Please also state the size of a mature cell.

Line 214: “Further, between the S-layer and the cell membrane is a pocket that is absent in the mature cell (Fig. 4a)”. It is unclear what is this ‘pocket’. The arrow in Fig. 4a appears to be pointing to a relatively empty region of the volume. Fig 4 needs significant revisions more generally: Are the preloops in fig 4e and 4f between the cell and the plug or within the interstitial spaces (between two plugs)? Also, I am unable to resolve a breakage in the sheath mark by an ‘*’ in Fig 4b.

Line 220: “similar thickness”. How thick?

In the discussion:

This needs considerable revision. First paragraph reads like part of a chapter from the early draft of a thesis or a review, and is almost entirely unrelated to the manuscript.

***Line 272: “the pre- β -hoop inserts into a temporary opening in the sheath layer between two cells”. What evidence is there that this opening is ‘temporary’? Can you rule out that this opening was not caused by damage during cryoEM grid preparation?

Line 275: These steps are based on the observations in tomograms (Fig. 4) and are consistent with prior observations of the assembly and translocation of highly ordered cellular protein arrays, and bacterial and eukaryotic viruses”. What prior observations?

Line 279: “...external to a bilayer membrane without transmembrane viral proteins”. This is unclear.

***Line 291-300: “Capable of coping with pressure approaching... This section of the discussion is highly speculative. What evidence is there that the “ β -hoops are well sealed” to gaseous exchange? How do you know what is the gas permeability of the sheath without some functional experiments?

***Line 300: “Our data supports the previous proposed model⁴² that the accumulating gas can stretch the size of the pores near the cleft region, increasing the permeability of sheath layer for gas exit”. What is the empirical evidence that the sheath is impermeable to gas? This seems highly speculative. What are the ‘pores’ in the structure? These have not been described other than an apparent breakage in only one of the tomograms in the dataset.

Line 308: “Accumulation of limiting nutrients”. No evidence of accumulating nutrients is presented in this manuscript, therefore this seems highly fanciful.

In the methods: Please describe how subtomogram averaging resolution was calculated in more detail. Was this gold standard FSC?

Reviewer #2 (Remarks to the Author):

Report for “An amyloid-like protein polymerizes into sheath during archaeal cell growth”

This manuscript by (Wang, Zhang et al.) uses an integrative approach by combining cryo-ET and AlphaFold predictions to reveal the structure of the proteinaceous sheath of the archaeon *Methanospirillum hungatei*. The authors show that the sheath is comprised of B-hoops, each hoop consisting of various number of rings (2, 3, 4, 5, or 6). They reveal that the sheath protein (SH) which

assembles to the rings (to subsequently form the hoop) has three major domains (a cap domain, an amyloid-like domain and a linker). Subsequently, they discuss in detail the various interactions that enable and stabilize the formation of the rings and the hoops by SH and compare these interactions with those present in amyloid fibrils (like alpha synuclein). Finally, they present a model of how this sheath might assemble in situ and discuss the benefits of such a non-crystalline sheath to *M. hungatei* by acting as a pressure regulator.

The manuscript is a robust piece of work that significantly enhances our understanding of the archaeal sheath. It is well-written and clearly illustrated and I support its publication in a prestigious journal like Nature Communications.

I have the following suggestions/questions to the authors:

- In the abstract, the authors write: "Some archaeal cells possess proteinaceous sheath and surface layers (S-layer) outside their lipid membranes, distinct from eukaryotic and bacterial cells, the former lack such layers and the latter have a non-proteinaceous peptidoglycan wall." Please note that this is true for Gram-positive bacteria. However, Gram-negative bacteria have an outer membrane surrounding the peptidoglycan layer. Also, some bacteria like *Caulobacter crescentus* have an S-layer.

This same comment applies to the first few sentences of the introduction where the authors say that "in Bacteria, that membrane is enclosed by a peptidoglycan wall".

- In figure 1C legend, "Blue arrowheads point to extra densities attached to the β -hoops near the plug ". Did the authors try to classify their particles and do averaging for the hoops with the extra density only? Probably this will resolve this density which the authors later speculate could be binding to the base loops and be related to either plug formation or cell division?

- In figure 1C (left panel, near the "end plug" region, just to the right of the blue boxed area), how do these hoops look like? Are they complete hoops with smaller diameter? Incomplete hoops?

- Line 140: "Also, cap-cap interactions likely reinforce this K136-D43 interaction" is "K136" a typo? Shouldn't it be "K363"?

- *csgA* is only mentioned in the legend of Figure 2 but never in the main text? Probably define it in the main text?

- Lines 161-162: "The docking also reveals possible sites of glycosylation, where amino acids Thr88, Ser144 and Asn167 in the cap domain are exposed to the exterior of the cell (Fig. 3b)." two points regarding this: 1) what is the reason the authors think that this Thr, Ser, and Asn are glycosylated? Can they cite a reference? 2) Please check the numbering of the amino acids again (in the text it is written Thr88 and Asn167 while in Figure 3b it is Thr86 and Asn157, please double check).

- Line 206 "In situ structure of a growing cell"

I found this part to be the weakest part of the manuscript for the following reasons:

a) From what I understand in the text, the authors have only one tomogram at this stage? Is this correct? At least they show the same cell in Figure 4 and Movie S4. If this is really the case (only one example), then I would be hesitant to build a model based on one tomogram only. Can the authors add at least one more example? They can just scan their grids and identify another growing cell (I assume

you can easily identify them with 2D projection images) and collect a tomogram there to see if they find a pocket and similar densities there as they describe in this example? If they can show one more tomogram(s) of a growing cell with the pocket and densities therein, and discontinuities in the sheath then this would be more convincing. Alternatively, if the authors cannot add another example for one reason or another, then I would make this part more tentative, it reads very strong in the paper now (it is highlighted in the abstract, a whole result section, a main figure, and in the discussion section) for something based on one example.

b) The model as stated by the authors raises some questions. For example, I assume the authors are imagining a pool of “pre B-hoops” (stage 3 in Figure 5b) with various rings (like 2, 3, 4, 5, etc), then how would a “pre B-hoop” with the exact same number of partial rings be incorporated into the growing hoop in stage 4? In other words, if the growing hoop (stage 4 in Figure 5b) is a 5-ring hoop, then why would a 5-ring pre B-hoop (stage 3 in Figure 5b) bind to it (from the assumed pool of pre-hoops) and not any other one with a different ring number? An alternative hypothetical model, for example, is that a whole ring can assemble first in the sheath and then another one builds on top of it and so on to build a hoop ultimately (I am mentioning this model just as an example of an alternative one, although this model would also have its own problems). As I mentioned above, if the authors can add another example to support their model then this would be more convincing, if not, then probably just make this part more tentative?

c) Just for accuracy, I would vary the number of the hoops in the model in Figure 5 b, c. The authors only show 4-ring hoops.

d) In movie S4, there appears to be a circular structure on the right side of the interstitial space (just to the left of the dark densities which the authors interpret as nascent SH polymers). Can the authors comment on that?

- Reference 43 is now published in structure1, I would cite the published paper instead of the bioRxiv version.

- I liked how the authors compare the sheath to other systems in the discussion (like gas vesicles, some viruses, etc..). However, I missed a comparison between the sheath and bacterial S-layer (see, for example reference2 or reference3 and references therein). I would suggest adding something in that regard to the discussion section.

- In the methods section, the authors mention that they imported the orientations and coordinates of particles from PEET to Relion4. I have not used Relion4 myself, but is this a straightforward thing? If not, probably elaborate more (and make any relevant scripts used to that end public if possible) for the benefit of the community?

- Finally, I would highlight in the main text that the directional resolution is anisotropic (when the resolution values are given), as the authors already show in Figure S1.

Congratulations on this nice work and Good luck!

Reference:

1. Dutka, P. et al. Structure of *Anabaena flos-aquae* gas vesicles revealed by cryo-ET. *Structure* 31, 518-528.e6 (2023).
2. Von Kügelgen, A. et al. In Situ Structure of an Intact Lipopolysaccharide-Bound Bacterial Surface Layer. *Cell* 180, 348-358.e15 (2020).
3. Bharat, T. A. M., Von Kügelgen, A. & Alva, V. Molecular Logic of Prokaryotic Surface Layer Structures. *Trends in Microbiology* 29, 405–415 (2021).

REVIEWER RESPONSE

Summary: We thank the two reviewers for the very careful review of our paper and the editor for the interest of publishing our work. As you will see from the following point-by-point responses to the reviewers' comments, we have attempted to address all the concerns raised by the reviewers and incorporated their suggestions in the revised figures and manuscript. In particular, we have clarified the number of tomograms we have obtained for the growing cell experiment and included additional example (Supplementary video 6). We have rewritten much of the Discussion to tone down the putative sheath genesis model from three aspects: First, we added the word "putative" whenever referring sheath genesis model; Second, we include other alternative model in addition to the original one as depicted in the new Figure 5; Third, we restructured our Discussion section to focus on direct structure observation and significantly shortened our text discussing the putative sheath genesis model. To facilitate your perusal of this statement, we have copied the reviewers' comments in **black** and our responses are shown in **blue**.

Reviewer #1:

Wang et al report a beautiful in situ structural characterization of an archaebacterial sheath protein with a modest number of tomograms (~12 tomograms). Sutomogram averaging to 7.9 Å resolution enabled them to resolve the domain organization of the sheath (cap, beta-sheet domain, and linker). The authors then use an alpha-fold predicted atomic model to perform additional interpretation of the organization of the sheath protein within their in-situ tomographic maps.

In one of the tomographic volumes, a smaller cell is observed that contains openings in the sheath near to the cell, an incomplete plug and some protein filaments. The authors speculate that these are intermediates of sheath assembly.

While the tomographic data is of a high quality, I am concerned that a several of the main points in the manuscript do not have sufficient evidence to support their conclusions:

We appreciate your very careful review and the constructive criticisms, which we considered very carefully and tried to address rigorously, as detailed in the replies to the itemized comments below.

1) Hydrogen bonds in the amyloid core of the protein (Figure 3).

The authors interrogate an alphafold model of SH and describe a hydrogen bonding network linking the two beta-sheets of the amyloid core of SH. This seems to be highly speculative, particularly given the known limitations of alphafold to predict such fine details of a protein. The authors must at the very least present evidence that their model does not contain atomic clashes and be more cautious in their description of this feature in the alphafold model.

We have deleted the interoperation of the hydrogen bonds, instead we just point out the observation that there is a cluster of aromatic residues in this region. The text (see lines 186-188), and Figure 3c are modified accordingly.

2) Disulphide bonds in the sheath (Figure 3).

Using the alphafold model of SH and placing this in their subtomogram map, the authors note that this places cysteine residues from neighbouring SH molecules 8-13 Å apart. They then assert that the linker is flexible and has formed a disulphide bond. Surely if the linker region had formed a disulphide bond, then it would no longer be as flexible and tomographic density from the one SH protein to its neighbour would be well-resolved? I wonder if the authors might consider providing biochemical evidence to support this assertion by blocking free thiols with iodoacetamide, then running an SDS-PAGE gel without a reducing agent and comparing it to an SDS-PAGE gel with a reducing agent. Presumably if the SH protein formed intermolecular disulphides, you would not see a monomer band (by immunoblot) in the absence of a reducing agent.

You are right, we think that once the putative disulfide bonds are formed, the structure will be rigid. While the suggested biochemical experiment is excellent, it cannot be performed at present since the SH protein does not enter SDS-PAGE gels to be resolved as clearly identifiable gel bands, as has long been recognized.

Upon careful reading of this comment and our original submission of this part of our text, we now recognize that the original writing is not very clear. Thus, we have carefully revised this part of the paper (see lines 213, 218-221). The main point we tried to answer the questions about how sheath accomplish a rigid and linear

architecture while there are clearly structural variations among β -hoops of different number of β -rings and across different β -rings within each β -hoop. The revised text also made clear that only intra-hoop β -rings contain densities corresponding to disulfide bonds.

3) Putative intermediate Pre-beta-hoop (Figure 4).

The authors discuss a single tomogram that includes a smaller cell than the others in their 10 tomogram dataset. In it they conclude that this is a 'growing cell' and identify filamentous material between the cell membrane and sheath that they describe as 'Pre-beta-loops'. This seems highly speculative. How can they be sure that this rod-shaped tomographic density is formed of SH and not some other extracellular protein? This must be much more cautiously stated. Perhaps this could be supported with more convincing evidence.

You have a great point. We have revised both the Result and Discussion sections to incorporate your point.

In Results, we now have toned down this claim by adding "putative" in front of pre-beta-hoop and provided an alternative interpretation of the filamentous density (see Fig. 4 legend and lines 280-282).

In the Discussion, we have significantly toned down the interpretation of the data and suggested alternative pathways as illustrated in Figure 5. We agree that the current data do not allow us to definitively establish that the rod-shaped density is hoop precursors. Nonetheless, we hope that the alternative pathways would stimulate future studies to resolve these interesting questions (see lines 326-354).

Our interpretation is based on the following considerations. The filamentous sheath precursor would be about the same approximate diameter of the mature hoop (~10 nm) which is similar to the *M. hungatei* flagellum and thus difficult to individually identify whereas the pili filament is ~6 nm. (Ref #4 and Fig. 1a, 5c). No other filament candidates are known at present nor are there other any secreted cell proteins detected thus far that would be candidates. We have indicated that it could be flagellum in Figure 5b, c. More importantly, the filamentous density appears to be uniformly curved similar to that of the mature sheath hoop (Fig. 4e).

Overall, the manuscript was also very difficult to follow, with numerous sentences that contain errors or were ambiguous/impossible to interpret. Some of the figures are also unclear with incomplete figure legends. I wonder if the first draft of the manuscript was written by the first author (there is certainly considerable enthusiasm, which I applaud!) but perhaps the lead author, who is highly experienced and a world leader in cryoEM, has not given this manuscript sufficient attention to getting it in a form that is ready for peer review.

We appreciate this criticism and have made efforts to improve our writing for the revised manuscript. Indeed, the first draft was written by the first author and the senior authors have now worked side-by-side at great length to improve the original writing.

Below are some specific examples (but by no means an exhaustive list) of errors/problems in the manuscript (I have highlighted the more significant problems with a ***):

In the abstract

Lines 2-4: The first sentence is unclear.

We have rewritten it (see lines 2-4).

Line 13: What does "azimuthally augmented" mean?

It means that the number of strands within the β sheet is increased around the circumference of the cylindrical cell by addition of more subunits. We now made it clearly by defining it at first appearance (see line 15).

***Lines 14-18: "Tomograms of dividing cells...". This sentence misrepresents the data in the manuscript. None of the tomograms showed evidence of a 'dividing cell'. Only one tomogram showed a smaller cell with holes in the sheath and filamentous material in the space between two plugs. No structural data showing 'insertion' of beta-hoops' was presented in the manuscript.

The tomogram shown in Figure 4 is indeed an example of a dividing cell. The sample was prepared from exponentially growing cell cultures where only these short cells are observed in contrast to cells in stationary phase cultures that contain mature cells of ~ 7 μ m in length (see line 47, ref #7). In addition, in this particular example, the plug is not fully formed as yet. We are happy to provide images of dividing cells in earlier stages of the division process prior to membrane septation and formation of newly synthesized plugs depicted in Figure 4.

In the introduction:

Line 21: “cells in Eukarya have a simple lipid membrane”. This is an over-simplification. What about plant cells?

We have modified the offending passage to now mention the plant wall as well as discussion on microbial cell envelopes (see lines 24-29).

Line 32: “500 um”. Is this dimension referring to length?

Yes, we have modified in the text (see line 41).

Line 37: “Its cylindrical shape is essential for optimizing gas exchange...”. No citation of the literature is given to backup this statement. This point also seems tangential to the results presented in the manuscript.

Rather, the cylindrical cell shape enhances gas exchange as the cell surface area is increased by about 2-fold over that of a spherical cell. We have replaced “essential” by “enhances” modified in text (see line 46).

Line 51: “Indeed, the seemingly unrelated characteristics...”. I did not understand this sentence. In particular, what are the unrelated characteristics?

The passage has been clarified by listing the characteristics (see lines 62-64).

Line 56: “What selective advantage does such an organization provide for cell growth and gas regulation?”. There are no experiments reported in this manuscript that address this question.

We have deleted this statement. In the revised Discussion, we mentioned the relevance of the sheath structure to gas regulation (see lines 373-388).

Line 60: “oligomerize azimuthally around the cylindrical cell into a ring structure”. This sentence is unclear.

By “oligomerize azimuthally”, we meant to convey that the number of SH subunits increases by addition of more subunits around the circumference of the cylindrical cell (see line 74).

Line 61: What is a “ β -arch kernel”?

We have rephrased the sentence to reference the concept of “ β -arch kernel”, which was first introduced in by Li et al, as cited as Ref #19 now, the previous citation *Chiti, F. et al 2006* has been removed now (see lines 74-76).

In the Results:

***Line 84: “explaining the failure of the early effort to resolve the sheath structure based on crystallinity.” I don't think one can be so definitive about the limitations of another groups work.

We and others have been working on the structure assuming crystallinity based on previously published diffraction spots (cited as ref #14) and no structures have been obtained. Particularly, we indicated that the failures including that of our own (see lines 98-99).

Line 88: “No long-range periodicity in the arrangement of different membered β -hoops is observed along the axial direction (Fig. 1c insets)”. What is long range periodicity? Is this referring to the distribution of different sized β -hoops and them not being arranged with a pattern? If yes, I don't think the information contained fig. 1c properly/thoroughly demonstrates this point.

This has been clarified in the main text (see lines 102-107).

Line 90: “sorted out”. What does mean?

We have changed “sorted out” to “classified” (see lines 109-110).

Line 96: “Repeat units appear”. What are the repeat units?

This has been clarified in the main text (see lines 116-118).

Line 107: “reveals”. The tense in much of manuscript jumps about from the past to the present tense. Please be consistent.

We did multiple rounds of careful proof-reading to ensure correct tense usage. The guiding principle we uphold is that all factual statements are in present tense (e.g., this structure shows) and all past actions are described in past tense (e.g., cells were cultured).

Line 108: “ β rings alternates along the axial axis”. What do the authors mean by ‘alternates’?

The sentence was rewritten to make it clear (see lines 129-130).

Line 113: “arch shape (top panel of Fig. 1j)”. I could not identify an arch shape in this figure. Please explain or improve this figure.

We have changed the word “arch” to “annulus sector”. Also, two arch shapes have been added in Fig. 1j to indicate the outer and inner arches of an annulus sector (see lines 134 and Fig. 1j).

Line 129: “SH can acquire more self-stability”. More self-stability than what? I am also a little unclear what is self-stability.

Line 131: “form more hydrophobic cores with participation of a large amount of aromatic amino acids”. More hydrophobic cores than what? Please explain. Also, what is a large amount of hydrophobic residues?

The sentence was rewritten to make it clear. Definitive words were being removed and replaced by suggest phrases (see lines 153-156). (A combined answer to the 2 comments above)

Line 136: “is two residues shorter at the fold area, which leaves notch”. I don’t understand this statement. I am also not sure that the word ‘notch’ makes much sense here. Line 137: “ β -strands forming a β - sheet normally only connect via the polypeptide backbone interactions”. I don’t understand what ‘connect’ means.

Line 140: “Also, cap-cap interactions likely reinforce this K136-D43 interaction”. It does not make biophysical sense to describe the distal cap-cap interaction as ‘reinforcing’ the K136-D43 interaction.

Thank you for the great catch! We agree with these all and rewrote much of this paragraph to fix the issues raised (see lines 160-173). (A combined answer to the 2 comments above)

Line 154: “In the cap domain, a 23Å-long helix in the AlphaFold-predicted model matches well...(Fig. 3b)”. In fig 3b, is this the helix described as the ‘longest’ in the figure legend or one of the other helices in the cap domain?

Yes, we have updated the figure legend to indicate that the ‘longest’ helix in SH model is 23Å-long.

***Line 157: “corresponding to the predicted locations of hydrogen bond core between the side chains of amino acids near the center (Fig. 3a red box and Fig. 3c).” This sentence is poorly written. What is a ‘hydrogen bond core’. Fig. 3c needs to be improved: the presence of hydrogen bonds is not clear. I suspect this might be over-interpretation of an alphafold model since usually the precision of Alphafold is not sufficient to reliably interrogate side-chain-side chain interactions. Have you performed an energy minimisation of the model and ensured that there are no steric clashes of sidechain atoms?

We have deleted the interoperation of the hydrogen bonds, instead we just point out the observation that there is a cluster of aromatic residues in this region (see lines 186-189 and Figure 3c).

Line 173: “locational relationship”. What does this mean?

We have replaced “locational” to “spatial” in the text (see line 204).

Line 176: “and following axial arrangement of SH”. This is unclear.

We have replaced “axial arrangement of SH” to “ β -ring stacking arrangement within a β -hoop” (see lines 208).

Line 194: “To answer this question, we compare the subtomogram 95 averages of 4- β -ring, 5- β -ring, and 3- β -ring hoop to investigate the local subunit conformational changes and inter- β -hoop interfaces (Fig. 3g)”. What do the ‘star’ symbols and dashed lines represent in Fig 3g?

We have deleted the stars in Figure 3g now. And the dashed line is explained in the Result (see lines 233-250).

Line 197: “connected by β -sheets domain”. This is unclear.

Line 198: “between adjacent β -hoops, the contact points at the caps between neighboring β -hoops seem to remain the same”. The same as what?

We have rewritten these sentences to clarify it in the text (see lines 233-241). (A combined answer to the 2 comments above)

Line 202: “Such flexibility would compensate for differences in the angles between adjacent β -hoops to maintain consistent molecular bonds between contacting caps, and allow all β -hoops to be aligned along the axial direction”. What is the evidence that there are consistent molecular bonds between contacting caps?

Great catch! We have changed the word “bonds” to “contact”. We can only see they contact each other, and have no evidence of bond formation, although the fact that the sheath always stays together suggests bond formation (see line 247-250).

Line 212: “we observed a growing cell measured around 480 nm long (Fig. 4a), much shorter than that of the mature cell (Fig. 1a)”. You have not presented any evidence that the cell is growing. Please also state the size of a mature cell.

Cells that used to prepare this figure (Fig 4a) were obtained from actively growing mid-log phase cultures. The “short” cells are not observed in stationary phase cultures whereas cells are uniform and much longer in length $\sim 7 \mu\text{m}$ (see line 47, ref #7). In addition, in this particular example (Fig 4a), the plug is not fully formed as yet.

Line 214: “Further, between the S-layer and the cell membrane is a pocket that is absent in the mature cell (Fig. 4a)”. It is unclear what is this ‘pocket’. The arrow in Fig. 4a appears to be pointing to relatively empty region of the volume. Fig 4 needs significant revisions more generally: Are the preloops in fig 4e and 4f between the cell and the plug or within the interstitial spaces (between two plugs)? Also, I am unable to resolve a breakage in the sheath mark by an ‘*’ in Fig 4b. (Answered separately)

- It is unclear what is this ‘pocket’. The arrow in Fig. 4a appears to be pointing to relatively empty region of the volume.

The “pocket” region is now more clearly indicated in Fig. 4a, b and encircled in red ellipses. It is defined as the enlarged space between the Cell Membrane (CM) and the S-layer and located near the cell corners where the S-layer becomes detached from CM to form the “pocket” or “bubble”.

- Fig 4 needs significant revisions more generally: “Are the preloops in fig 4e and 4f between the cell and the plug or within the interstitial spaces (between two plugs)?”

Yes, the pre-hoops shown in Fig. 4e are within the interstitial space, and 4f shows only the sheath with opening at the same region as 4e. The dotted boxes in Fig. 4d indicate the regions of structure shown in Fig. 4e. We have updated Fig. 4d-f and the accompanying legend to clarify.

- “Also, I am unable to resolve a breakage in the sheath mark by an ‘*’ in Fig 4b.”

We have indicated the breakage using a yellow circle in Fig. 4b now. The breakage is the empty space between the incomplete plug and the sheath layer.

Line 220: “similar thickness”. How thick?

The thickness is around 10nm. We have added that in the text and Fig. 4 (see line 266 and Fig. 4e).

In the discussion:

This needs considerable revision. First paragraph reads like part of a chapter from the early draft of a thesis or a review, and is almost entirely unrelated to the manuscript.

We appreciate your criticism and have rewritten much of this section. The revised first paragraph set the stage for the following discussion points, which are all related to the observations reported in the current paper (see lines 284-306). This way of presentation is indeed different from the common practice of having the first paragraph summarizing the findings, which are in our second paragraph in Discussion (see lines 307-325).

***Line 272: “the pre- β -hoop inserts into a temporary opening in the sheath layer between two cells”. What evidence is there that this opening is ‘temporary’? Can you rule out that this opening was not caused by damage during cryoEM grid preparation

We don’t have direct evidence of temporal progress for the temporary nature of opening and now acknowledge this important limitation at both the beginning (see lines 326-329) and the end (see lines 351-354) of the paragraph discussing our proposed model. Due to intrinsic limitation of cryoET, only frozen snapshots can be captured, and no direct temporal evidence can be obtained from frozen hydrated sample of dividing cells. We

also cannot eliminate the possibility of cryoEM sample damage, though given the gentle methods used here, it would seem unlikely because these sheath openings have not been observed in stationary phase cells.

Line 275: These steps are based on the observations in tomograms (Fig. 4) and are consistent with prior observations of the assembly and translocation of highly ordered cellular protein arrays, and bacterial and eukaryotic viruses". What prior observations?

Line 279: "...external to a bilayer membrane without transmembrane viral proteins". This is unclear.

The two statements in the original text are related to the same topic. We now have rewritten them to make the original idea clear. The prior observations refer to Ref #42 (see lines 344-351). (A combined answer to the 2 comments above)

***Line 291-300: "Capable of coping with pressure approaching... This section of the discussion is highly speculative. What evidence is there that the "β-hoops are well sealed" to gaseous exchange? How do you know what is the gas permeability of the sheath without some functional experiments?

***Line 300: "Our data supports the previous proposed model⁴² that the accumulating gas can stretch the size of the pores near the cleft region, increasing the permeability of sheath layer for gas exit". What is the empirical evidence that the sheath is impermeable to gas? This seems highly speculative. What are the 'pores' in the structure? These have not been described other than an apparent breakage in only one of the tomograms in the dataset.

We agree and have now deleted this speculative statement in the revised manuscript. Our original writing was not clear on this, and we have rewritten this passage as follows (lines 373-388) with added new Extended Figure 6b to illustrate the distinctive surface property. At the same threshold level, the inter-β-hoop interface shows more 'holes' (indicated by red arrowheads in Fig. 1j, which is what we meant by 'pores' previously) compared with the intra-β-hoop interior surface (Fig. 1j middle panel). (A combined answer to the 2 comments above)

"M. hungatei sheath has the extra-ordinary capability of resisting force of ~300 atmospheric pressure¹⁷, yet should allow exchange of methane, hydrogen and water to support basic intracellular functions. Notably, buoyancy-control gas vesicles within bacterial and archaeal cells are also thin-walled cylinders with distinctive hydrophilic exterior and hydrophobic interior assembled from a single β-hairpin-containing protein^{43,44}, thus differ from the archaeal sheath which fashions alternating hydrophobic/hydrophilic patches (Extended Data Fig. 6). Intra-hoop SH interactions are extensive, explaining its capability to resist large force. The interior of each β-hoop has no holes and thus is well sealed by its amyloid-like β-sheet domains (Fig. 1e and j), therefore the wedge-shaped cleft between β-hoops, which varies in shape and measures up to ~30° in angle and 40 Å in width (Fig. 1j), appears to be the only outlet for waste gas."

Line 308: "Accumulation of limiting nutrients". No evidence of accumulating nutrients is presented in this manuscript, therefore this seems highly fanciful.

We agree and have deleted this phrase.

In the methods: Please describe how sub-tomo averaging resolution was calculated in more detail. Was this gold standard FSC?

Yes, it was gold standard FSC. We have indicated it in the Method section (see lines 464-466).

Reviewer #2:

Report for “An amyloid-like protein polymerizes into sheath during archaeal cell growth”

This manuscript by (Wang, Zhang et al.) uses an integrative approach by combining cryo-ET and AlphaFold predictions to reveal the structure of the proteinaceous sheath of the archaeon *Methanospirillum hungatei*. The authors show that the sheath is comprised of B-hoops, each hoop consisting of various number of rings (2, 3, 4, 5, or 6). They reveal that the sheath protein (SH) which assembles to the rings (to subsequently form the hoop) has three major domains (a cap domain, an amyloid-like domain and a linker). Subsequently, they discuss in detail the various interactions that enable and stabilize the formation of the rings and the hoops by SH and compare these interactions with those present in amyloid fibrils (like alpha synuclein). Finally, they present a model of how this sheath might assemble in situ and discuss the benefits of such a non-crystalline sheath to *M. hungatei* by acting as a pressure regulator.

The manuscript is a robust piece of work that significantly enhances our understanding of the archaeal sheath. It is well-written and clearly illustrated and I support its publication in a prestigious journal like Nature Communications.

Thank you for your support!

I have the following suggestions/questions to the authors:

- In the abstract, the authors write:” Some archaeal cells possess proteinaceous sheath and surface layers (S-layer) outside their lipid membranes, distinct from eukaryotic and bacterial cells, the former lack such layers and the latter have a non-proteinaceous peptidoglycan wall.” Please note that this is true for Gram-positive bacteria. However, Gram-negative bacteria have an outer membrane surrounding the peptidoglycan layer. Also, some bacteria like *Caulobacter crescentus* have an S-layer.

This same comment applies to the first few sentences of the introduction where the authors say that “in Bacteria, that membrane is enclosed by a peptidoglycan wall”.

We have revised the first sentence in the abstract to incorporate your correction. The 2-5 sentences of the first paragraph were also rewritten to clarify this distinction (see lines 24-32).

- In figure 1C legend, “Blue arrowheads point to extra densities attached to the β -hoops near the plug “. Did the authors try to classify their particles and do averaging for the hoops with the extra density only? Probably this will resolve this density which the authors later speculate could be binding to the base loops and be related to either plug formation or cell division?

We did not include the particles near the plug region for sub-tomogram average, so we did perform classification. Now, we have picked new particles near that region, and the averaged structure shows the extra densities continuously attached on both lower edges of the β -hoops. Although the resolution is relatively lower because of limited particle numbers and preferred orientation issue. We are happy to provide this data for the reviewer on request and to incorporate it into the report.

- In figure 1C (left panel, near the “end plug” region, just to the right of the blue boxed area), how do these hoops look like? Are they complete hoops with smaller diameter? Incomplete hoops?

Regarding the “end-plug” question, this issue is unresolved at present and future studies are planned to address this interesting structure. We believe there may be additional protein present that provides the hook-like appearance.

- Line 140: “Also, cap-cap interactions likely reinforce this K136-D43 interaction” is “K136” a typo? Shouldn't it be “K363”?

We have corrected the text to K363 (see line 165 and 168).

- csgA is only mentioned in the legend of Figure 2 but never in the main text? Probably define it in the main text?

We have mentioned the α -synuclein and csgA bacterial curli protein in the text now (see lines 151-152) as examples of amyloid structures.

- Lines 161-162:” The docking also reveals possible sites of glycosylation, where amino acids Thr88, Ser144

and Asn167 in the cap domain are exposed to the exterior of the cell (Fig. 3b).” two points regarding this: 1) what is the reason the authors think that this Thr, Ser, and Asn are glycosylated? Can they cite a reference? 2) Please check the numbering of the amino acids again (in the text it is written Thr88 and Asn167 while in Figure 3b it is Thr86 and Asn157, please double check).

Glycan sites: Indeed, the sites are speculative predictions, but based on our prior structure of *M. hungatei* flagella at about 3Å resolution (ref #4) that demonstrate O-linked glycosylation in its flagella, the two Ser and Thr exposed residues would be likely candidates for similar modification. The Asn157 would be a surface exposed and potential N-linked glycosylation site. We now cite the flagella study (see line 192).

Thank you for the great catch! We also correct the Thr88 to Thr86, and Asn167 to Asn157 in the text (see lines 192-193).

- Line 206 “In situ structure of a growing cell”

I found this part to be the weakest part of the manuscript for the following reasons:

a) From what I understand in the text, the authors have only one tomogram at this stage? Is this correct? At least they show the same cell in Figure 4 and Movie S4. If this is really the case (only one example), then I would be hesitant to build a model based on one tomogram only. Can the authors add at least one more example? They can just scan their grids and identify another growing cell (I assume you can easily identify them with 2D projection images) and collect a tomogram there to see if they find a pocket and similar densities there as they describe in this example? If they can show one more tomogram(s) of a growing cell with the pocket and densities therein, and discontinuities in the sheath then this would be more convincing. Alternatively, if the authors cannot add another example for one reason or another, then I would make this part more tentative, it reads very strong in the paper now (it is highlighted in the abstract, a whole result section, a main figure, and in the discussion section) for something based on one example.

Our experiment of dividing cells consisted of 37 tomograms. One more example is now included in the new movie (Supplementary Video 6). Nonetheless, your point is well taken, and we have toned down the model and now have included alternative pathway/interpretation of the observed filamentous densities (see lines 280-282, 326-354 and Fig. 5 updates).

b) The model as stated by the authors raises some questions. For example, I assume the authors are imagining a pool of “pre B-hoops” (stage 3 in Figure 5b) with various rings (like 2, 3, 4, 5, etc), then how would a “pre B-hoop” with the exact same number of partial rings be incorporated into the growing hoop in stage 4? In other words, if the growing hoop (stage 4 in Figure 5b) is a 5-ring hoop, then why would a 5-ring pre B-hoop (stage 3 in Figure 5b) bind to it (from the assumed pool of pre-hoops) and not any other one with a different ring number? An alternative hypothetical model, for example, is that a whole ring can assemble first in the sheath and then another one builds on top of it and so on to build a hoop ultimately (I am mentioning this model just as an example of an alternative one, although this model would also have its own problems). As I mentioned above, if the authors can add another example to support their model then this would be more convincing, if not, then probably just make this part more tentative?

We have revised this paragraph to make it as a provisional model as suggested (see lines 326-354). These are all excellent questions! Like many other cellular cryoET data, as the first attempt to explore the complex question of cell division, our observations raised more questions than answers. We recognize that at this time, we don't have data to inform the details of steps and accordingly, we have revised the description of the proposed model by deleting the details about how pre-β-hoops and how the full β-hoops are assembled. Specifically regarding to the question about how the hoop is assembled, we have added a new figure (Extended Data Fig. 2) showing that hoops could have different numbers of rings within the same hoop (i.e., the ring counts in a hoop clearly change), suggesting that it is possible (although it does not rule out switching of rings post hoop assembly) that partial hoops could be assembled first prior to being inserted.

c) Just for accuracy, I would vary the number of the hoops in the model in Figure 5 b, c. The authors only show 4-ring hoops.

This variation has been added into Figure 5b, c now.

d) In movie S4, there appears to be a circular structure on the right side of the interstitial space (just to the left of the dark densities which the authors interpret as nascent SH polymers). Can the authors comment on that?

The circular appearance of these densities is an illusion of the same putative pre- β -hoop density due to oblique sectional view.

- Reference 43 is now published in structure 1, I would cite the published paper instead of the bioRxiv version.

This reference has been updated as new cited ref #43, also we added another new published paper related to high resolution Gas Vesicle structure as cited ref #44 (Huber, S. et al. 2023)

- I liked how the authors compare the sheath to other systems in the discussion (like gas vesicles, some viruses, etc.). However, I missed a comparison between the sheath and bacterial S-layer (see, for example reference2 or reference3 and references therein). I would suggest adding something in that regard to the discussion section.

Following your suggestion, we have added a statement about how archaeal sheath compares to S-layer in both archaea and bacteria. Functionally, the archaeal sheath plays similar roles as bacterial peptidoglycan or murein -- preserve cell integrity by withstanding the turgor and contributes to the maintenance of a defined cell shape. In bacteria, it also serves as a scaffold for anchoring other cell envelope components.

<https://www.cell.com/trends/microbiology/fulltext/S0966-842X%2820%2930258-4>

Legend: Schematic diagrams of prokaryotic cell surfaces are provided to illustrate the variety of anchoring mechanisms used by S-layers to assemble on cells. (A) Representation of the archaeal cell surface using the crenarchaeon *Sulfolobus islandicus* as an example, experimentally described using high-resolution electron cryotomography (cryo-ET) [23.]. Long stalk-like densities of the S-layer are buried within the outer membrane. (B) A generic Gram-positive bacterial outer surface with the S-layer buried within the cell wall. (C) The envelope of the Gram-negative bacterium *Caulobacter crescentus*, as described using cryo-ET [18.]. The anchoring domain of the S-layer is noncovalently attached to the O-antigen of LPS. Abbreviations: IM, inner membrane; LPS, lipopolysaccharide; PG, peptidoglycan; OM, outer membrane.

The archaeal S-layer and sheath structures are distinct from each other. The former lacks the structural rigidity of sheath as evidenced: for example, osmotically shocked cells show detachment of S-layer from the sheath but not from cell membrane. In contrast, the *M. hungatei* sheath is exceedingly rigid. Treatment with harsh agents or when sonicated, it retains the characteristics of sheath size and shape!

All archaea cell types except those with pseudo-murein walls have S-layers of one kind or another. They are all spherical to “partly deflated soccer balls” in shape. The S-layer cannot apparently confer cell shape. Also, cells with a sheath layer always have in addition an S-layer, indicating that sheath never replaces the S-layer.

- In the methods section, the authors mention that they imported the orientations and coordinates of particles from PEET to Relion4. I have not used Relion4 myself, but is this a straightforward thing? If not, probably elaborate more (and make any relevant scripts used to that end public if possible) for the benefit of the community?

Details have been elaborated in the Method now (see lines 453-458).

- Finally, I would highlight in the main text that the directional resolution is anisotropic (when the resolution values are given), as the authors already show in Figure S1.

We have added this information in the main text (see lines 111-112).

Congratulations on this nice work and Good luck!

Thank you for your suggestions and support!

Reference:

1. Dutka, P. et al. Structure of *Anabaena flos-aquae* gas vesicles revealed by cryo-ET. *Structure* 31, 518-528.e6 (2023).
2. Von Kügelgen, A. et al. In Situ Structure of an Intact Lipopolysaccharide-Bound Bacterial Surface Layer. *Cell* 180, 348-358.e15 (2020).
3. Bharat, T. A. M., Von Kügelgen, A. & Alva, V. Molecular Logic of Prokaryotic Surface Layer Structures. *Trends in Microbiology* 29, 405–415 (2021).

REVIEWERS' COMMENTS

Reviewer #1 (Remarks to the Author):

This is a much-improved manuscript. I recommend publication in Nature Communications. Below I have suggested a few minor revisions that the authors could consider.

Minor comments:

I accept all of the authors points on 'growing cell' but I am still not sure it is correct to say 'growing cell'. I would only say growing if the technique used to image the cell could show it was changing size, which they authors acknowledge is not possible with cryoET snapshots. You could consider 'immature cell' to overcome this issue.

Line 15: Authors changed "azimuthally augmented" to "azimuthally (ie. around circumference) augmented" in the abstract. I think this is still confusing because it is unclear what is the azimuth in the context of a cell. Also "augmented" is a bit cryptic
I suggest the authors consider plain language to explain what is after all a rather simple concept eg. "two giant beta-sheets composed of ~450 SH monomers that entirely encircle the outer circumference of the cell".

Lines 102-103: This might need further revision. Could write "as indicated by the arrangement of...". Saying 'pattern' is probably confusing because I think you are actually saying there is no pattern (ie. no periodicity). Also, I am also not sure it is necessary to quote the string of numbers in quotations.

Lines 160-163: This might need further revision. I didn't understand this. The authors could consider rewording this.

Lines 169-170: Minor grammatical error (missing indefinite article).

Lines 204-205: Minor grammatical error.

Throughout: There should be spaces between values and their units.

Line 235-241: This might need further revision. I did not understand this sentence – it is very long!

Line 246: Missing indefinite article ('a') on this line.

Lines 281-282: Minor grammatical error.

Line 309: Missing indefinite article ('a') on this line.

Lines 388-395: Very long sentence. I got a little lost. The authors could consider rewording this.

Reviewer #2 (Remarks to the Author):

The authors have satisfactorily answered all my questions and I recommend this paper for publication.

One final note: Probably add the low-resolution subtomogram average of the B-hoop particles with extra density (blue arrows in Fig. 1C) as a supplementary figure?

Congratulations!

REVIEWER RESPONSE

Summary: We have incorporated all suggestions made by the reviewers. Thank you for your suggestions and acceptance of our revised paper. To facilitate your perusal of this statement, we have copied the reviewers' comments in **black** and our responses are shown in **blue**. Note that the line numbers used below are referred in the Word file with "All Markup" enabled.

Reviewer #1 (Remarks to the Author):

This is a much-improved manuscript. I recommend publication in Nature Communications. Below I have suggested a few minor revisions that the authors could consider.

Minor comments:

I accept all of the authors points on 'growing cell' but I am still not sure it is correct to say 'growing cell'. I would only say growing if the technique used to image the cell could show it was changing size, which they authors acknowledge is not possible with cryoET snapshots. You could consider 'immature cell' to overcome this issue.

Response: thanks for pointing this out. We have modified the "growing" to "immature" as suggested, see lines 14, 18, 70, 235, 241, 247-248, Figure legend 4 & 5, and Movie legend 4 & 5.

Line 15: Authors changed "azimuthally augmented" to "azimuthally (ie. around circumference) augmented" in the abstract. I think this is still confusing because it is unclear what is the azimuth in the context of a cell. Also "augmented" is a bit cryptic

I suggest the authors consider plain language to explain what is afterall a rather simple concept eg. "two giant beta-sheets composed of ~450 SH monomers that entirely encircle the outer circumference of the cell".

Response: we have modified related text as suggested and replaced the "azimuthally" with "circumferentially", see lines 11-13, 65-66, 111-112, 120, 403, 405-406, 412, and Figure legend of 1f, 2e.

Lines 102-103: This might need further revision. Could write "as indicated by the arrangement of...". Saying 'pattern' is probably confusing because I think you are actually saying there is no pattern (ie. no periodicity). Also, I am also not sure it is necessary to quote the string of numbers in quotations.

Response: we have addressed this as suggested, the sentence now reads as follows: "As indicated by the arrangement of different membered β -hoops, no long-range periodicity is observed along the axial direction.", see lines 97-100.

Lines 160-163: This might need further revision. I didn't understand this. The authors could consider rewording this.

Response: indeed, there was a typo in the original sentence. We now have corrected "connection" to "connecting". We have rewritten this, see lines 153.

Lines 169-170: Minor grammatical error (missing indefinite article).

Response: we have fixed this, see line 159.

Lines 204-205: Minor grammatical error.

Response: we have fixed this, see lines 192.

Throughout: There should be spaces between values and their units.

Response: we have examined through the manuscript and added the spaces between values and their units.

Line 235-241: This might need further revision. I did not understand this sentence – it is very long!

Response: we have rewritten the related paragraph to make the point clear, the sentence now reads as follows: "As shown in Figure 3g, the only contact point between neighboring β -hoops is at the cap domains of

their outermost β -rings and remains the same; but the angle formed between them (i.e., the angle formed by red and pink dashed lines in Fig. 3g) is larger for β -hoops containing more β -rings.”, see lines 219-226.

Line 246: Missing indefinite article ('a') on this line.

Response: we have fixed this, see line 231.

Lines 281-282: Minor grammatical error.

Response: we have fixed this, see lines 266.

Line 309: Missing indefinite article ('a') on this line.

Response: we have fixed this, see line 288.

Lines 388-395: Very long sentence. I got a little lost. The authors could consider rewording this.

Response: we now break this statement into two sentences, see lines 347-353.

Reviewer #2 (Remarks to the Author):

The authors have satisfactorily answered all my questions and I recommend this paper for publication.

One final note: Probably add the low-resolution subtomogram average of the B-hoop particles with extra density (blue arrows in Fig. 1C) as a supplementary figure?

Response: we have added it as new Supplementary Fig. 4b.

Congratulations!